# Scaling Inference-Efficient Language Models

**Song Bian** [1] [*]   **Minghao Yan** [1] [*]   **Shivaram Venkataraman** [1]

## Abstract

Scaling laws are powerful tools to predict the performance of large language models. However, current scaling laws fall short of accounting for inference costs. In this work, we first show that model architecture affects inference latency, where models of the same size can have up to $3.5\times$ difference in latency. To tackle this challenge, we modify the Chinchilla scaling laws to co-optimize the model parameter count, the number of training tokens, and the model architecture. Due to the reason that models of similar training loss exhibit gaps in downstream evaluation, we also propose a novel method to train inference-efficient models based on the revised scaling laws. We perform extensive empirical studies to fit and evaluate our inference-aware scaling laws. We vary model parameters from 80M to 1B, training tokens from 1.6B to 30B, and model shapes, training 63 models. Guided by our inference-efficient scaling law and model selection method, we release the Morph-1B model, which improves inference latency by $1.8\times$ while maintaining accuracy on downstream tasks compared to open-source models, pushing the Pareto frontier of accuracy-latency tradeoff. Notably, our experiments reveal that wider and shallower models can yield efficiency gains while preserving accuracy.

## 1. Introduction

Scaling laws have shown immense value in guiding the development of large language models (LLMs) by establishing predictable relationships between model size, training compute, and performance metrics, such as loss and downstream tasks performance (Kaplan et al., 2020; Hoffmann et al., 2022; Muennighoff et al., 2023; Gadre et al., 2024). They reliably reduce the cost of training LLMs and improve

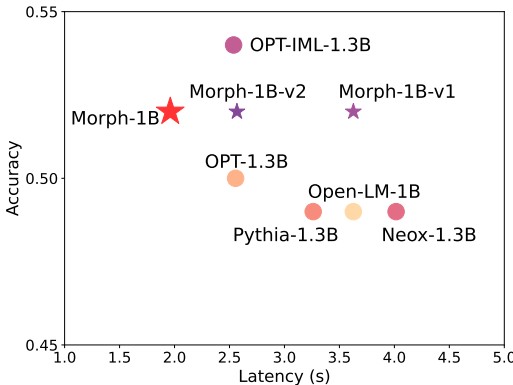

*Figure 1.* We train Morph-1B and its variant models on 30B tokens. The results indicate that Morph-1B maintains high accuracy on downstream tasks and achieves faster inference than open-source models and their variants. OPT-IML-1.3B achieves slightly higher performance on downstream tasks than Morph-1B since it is trained on 180B tokens (Iyer et al., 2022) and is instruction-tuned. We obtain the accuracy by evaluating models on 11 downstream tasks used by Open-LM (Gururangan et al., 2023). The inference latency is collected by using the Hugging Face `generate` function on a single NVIDIA Ampere 40GB A100 GPU with batch size 1, input length 128, and output length 256.

model design efficiency by accurately estimating an LLM's performance via the results of smaller language models, which can be developed using far less cost and fewer computing resources.

However, as the field progresses, it is increasingly evident that focusing solely on training does not adequately address the practical realities of deploying these models at scale (Touvron et al., 2023a). A key limitation of existing scaling laws is their disregard for inference costs, which dominate the long-term expenses of utilizing large models in real-world applications (Sardana et al., 2023). In other words, while compute-optimal models minimize training loss per unit of compute, they may result in models that are more expensive to serve, especially in latency-sensitive applications such as chatbots. The growing adoption of LLMs in reasoning systems also highlights the need for scaling frameworks that explicitly account for inference costs (Snell et al., 2024; Brown et al., 2024; Luo et al., 2024; Qi et al., 2024; Guan et al., 2025).

---
[*]Equal contribution [1]Department of Computer Sciences, University of Wisconsin-Madison, Madison WI, USA. Correspondence to: Song Bian <sbian8@wisc.edu>.

*Proceedings of the $42^{nd}$ International Conference on Machine Learning*, Vancouver, Canada. PMLR 267, 2025. Copyright 2025 by the author(s).

While a recent study (Sardana et al., 2023) has introduced scaling laws that consider the total number of FLOPS for training and inference, their constraint requires estimating the number of tokens inferred during the model's lifespan. As inference is performed repeatedly throughout a model's lifecycle, their scaling law (Sardana et al., 2023) is not practical for real-world applications.

In addition, current scaling laws focus on balancing model size (number of parameters) and the number of training tokens within a fixed compute budget[1] (Hoffmann et al., 2022; Muennighoff et al., 2023; Sardana et al., 2023; Gadre et al., 2024). Among these, the Chinchilla scaling law (Hoffmann et al., 2022) is the most renowned, demonstrating that the optimal training solution is $D = 20N$ for a fixed FLOPs budget, where $N$ is the number of parameters and $D$ is the number of tokens for training. However, in practice, we see that FLOPs are not a primary constraint. Models are trained for durations much larger than Chinchilla optimal (e.g., 1T tokens for Llama-7B and 8T tokens for Gemma-2-9B (Touvron et al., 2023a; Team et al., 2024b)). Additionally, practitioners choose the model size (number of parameters) based on the memory capabilities of the deployment device (Hu et al., 2024; Yao et al., 2024). Thus, we need scaling laws that can explicitly consider data size, device memory, and inference latency.

In this work, we aim to address the following question:

> *Given dataset and parameter constraints, can we train an inference-efficient and accurate model for downstream tasks?*

We first show that the number of parameters is not the exclusive factor affecting inference efficiency. As illustrated in Figure 2, the model architecture also plays a critical role. Following this observation, we introduce inference-efficient scaling laws, building upon the Chinchilla scaling law and incorporating model architecture considerations. Additionally, due to the disparity between model loss and accuracy in downstream tasks, we develop a novel method (Figure 6) that utilizes inference-efficient scaling laws to rank various model architectural choices. Our findings suggest that the relative ranking of loss predictions from scaling laws is more significant than their absolute values (§2).

To fit the inference-efficient scaling laws, we train more than 60 models ranging from 80 million to 339 million parameters for up to 13 billion tokens and record the loss of models. We also train several models with more than 1 billion parameters and 20 billion tokens to evaluate the predictive power of the fitted inference-efficient scaling

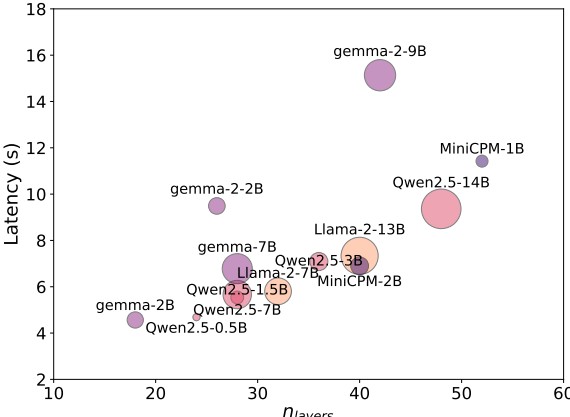

*Figure 2.* **Open-Source LLM's Inference Latency:** An overview of inference latency in open-source LLMs. The evaluated models include LLaMA (Touvron et al., 2023b), Qwen (Yang et al., 2024), Gemma (Team et al., 2024a;b), and MiniCPM (Hu et al., 2024). All evaluations were performed using the Hugging Face `generate` function on a single NVIDIA Ampere 40GB A100 GPU with batch size 1, input length 128, and output length 256.

laws. We observe that overtraining plays a critical role in obtaining an accurate scaling law and that our inference-efficient scaling law is more accurate and robust than the Chinchilla scaling law. Using only 6 data points and 85 A100 GPU hours for curve fitting, our inference-efficient scaling law can still accurately predict the loss of scaled-up models (§3, §4).

Lastly, we train the Morph-1B[2] model using the best model configuration predicted by our inference-efficient scaling law and ranking algorithm. Figure 1 summarizes our main results. Compared to other open source models of similar size, Morph-1B improves the inference latency by $1.8\times$ while maintaining accuracy over downstream tasks. These findings underscore the effectiveness of our inference-efficient scaling law. By designing a general scaling law that focuses on inference latency, our work can also capture the accuracy-efficiency trade-off for recent and future architectural optimizations, such as GQA (Touvron et al., 2023b; Dubey et al., 2024) and MLA (Liu et al., 2024a).

## 2. Scaling Laws

In this section, we first present the formulation of existing language model scaling laws in §2.1. Next, we introduce a scaling law for inference efficiency that takes into account the number of parameters, training tokens, and model shape in §2.2. Finally, we present a novel method to select

---

[1]The compute cost is approximated as $\text{FLOPs}(N, D) \approx 6ND$, where $N$ is the number of parameters and $D$ is the number of training tokens.

[2]The training code is available at https://github.com/Waterpine/open-lm-morph. The Morph-1B model checkpoint is available at https://huggingface.co/NaiveUser/morph-1b.

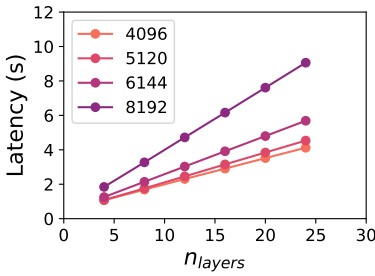
(a) Vary layers ($n_{\text{layers}}$), fix hidden size

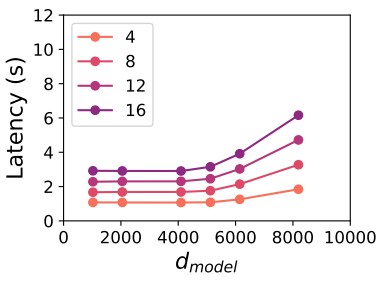
(b) Vary hidden size ($d_{\text{model}}$), fix layers

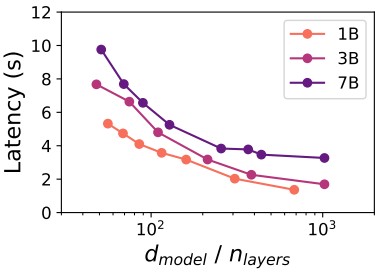
(c) Vary ratio ($d_{\text{model}}/n_{\text{layers}}$), fix size $N$

*Figure 3.* **Model Shape on End-to-End Inference Latency:** (Left) We illustrate the correlation between inference latency and the number of layers, with constant hidden size. Due to the sequential nature of LLM execution, latency increases linearly with the number of layers. (Center) We plot the relationship between inference latency and hidden size with the number of layers fixed. We see that model width does not affect latency for smaller models but only for larger models. (Right) We show the relationship between inference latency and aspect ratio, with the number of model parameters fixed. We see a downward trend in inference latency as we make the model wider and shallower. All evaluations were performed using the Hugging Face generate function on a single NVIDIA Ampere 40GB A100 GPU with batch size 1, input length 128, and output length 256.

inference-efficient language models for training using our scaling laws in §2.3.

## 2.1. Preliminaries

Scaling laws predict a model's loss based on the allocated compute resource $C$. Following OpenAI (Kaplan et al., 2020) and Chinchilla (Hoffmann et al., 2022), the compute resource $C$ is a function dependent on the model size $N$ and the number of training tokens $D$. The goal is to minimize model loss within the constraints of the available compute resources:

$$\arg\min_{N,D} L(N, D) \text{ s.t. FLOPs}(N, D) = C \qquad (1)$$

Using the formulation above, several scaling laws have been established (Kaplan et al., 2020; Hoffmann et al., 2022; Muennighoff et al., 2023; Sardana et al., 2023) to accurately model the performance of large language models from training a series of much smaller ones. The Chinchilla loss function $L(N, D)^3$ is widely adopted to predict a model's training loss:

$$L(N, D) = E + AN^{-\alpha} + BD^{-\beta} \qquad (2)$$

where $N$ is the number of parameters, $D$ is the number of tokens used for training and $A, B, E, \alpha, \beta$ are parameters to be learned. Through training multiple models and curve fitting, Chinchilla (Hoffmann et al., 2022) identify $D \approx 20N$ as the compute-optimal solution for large language model pretraining.

---

[3]Like Chinchilla (Hoffmann et al., 2022), we use smoothed training loss to estimate test loss.

## 2.2. Inference-Efficient Scaling Laws

Despite its popularity, the Chinchilla scaling law fails to resolve the following challenges:

- The FLOPs constraint outlined in Eq. (1) does not reflect how model training decisions are made in practice. First, both the model size and the training corpus are determined in advance to accommodate for resource constraints when deploying these models (Touvron et al., 2023a). Therefore, for each model and training corpus pair, training FLOPs is essentially a fixed constant (assuming training epochs are also predetermined). Furthermore, while the Chinchilla scaling law suggests training a 10B parameter model with 200B tokens, overtraining frequently occurs in practice. For example, the LLaMA-3-8B model uses 15 trillion tokens for training (Touvron et al., 2023a), while the Gemma-2-9B model utilizes 8 trillion tokens (Team et al., 2024b). These numbers are 44-93x larger than the Chinchilla optimal recommendation.

- Existing scaling laws focus only on how the number of parameters affects inference latency. However, as depicted in Figure 2, smaller models can sometimes exhibit higher inference latencies than larger models. For instance, MiniCPM-1B (Hu et al., 2024) has a higher latency compared to Qwen2.5-14B (Yang et al., 2024).

In view of this, we propose rewriting Eq. (1) as below to meet practical requirements:

$$\arg\min_{N,D} L(N, D) \text{ s.t. } N \leq N_C, D \leq D_C, T_{\text{inf}} \leq T_C \quad (3)$$

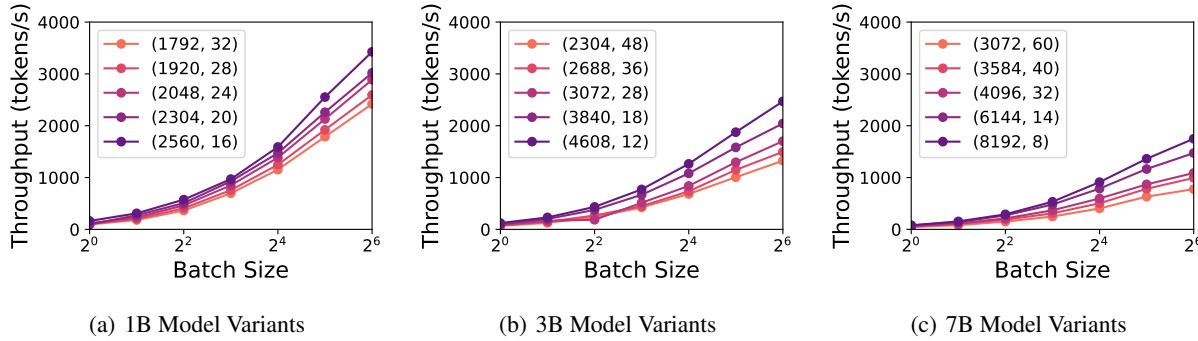

(a) 1B Model Variants      (b) 3B Model Variants      (c) 7B Model Variants

*Figure 4.* **Model Shape on Throughput:** We examine the relationship between inference throughput and model architecture by fixing the total parameter count and varying the hidden size and number of layers. Across different batch sizes, wider and shallower models consistently yield better inference throughput for large language models. Each tuple in the legend represents a model configuration: the first number is the hidden size $d_{\mathrm{model}}$, and the second is the number of layers $n_{\mathrm{layers}}$. All evaluations were performed using the Hugging Face generate function on a single NVIDIA Ampere 40GB A100 GPU with input length 128, and output length 256.

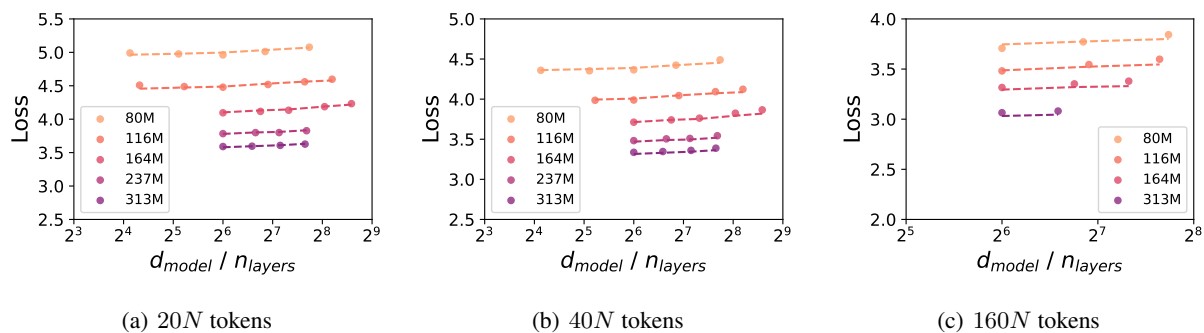

(a) $20N$ tokens      (b) $40N$ tokens      (c) $160N$ tokens

*Figure 5.* **Inference-Efficient Scaling Laws:** In this plot, each data point represents a training run with the given configuration. The dashed lines represent predictions based on the inference-efficient scaling laws outlined in Eq. (4). (Left) The number of training tokens is $20N$; (Center) The number of training tokens is $40N$; (Right) The number of tokens used for training is $160N$, where $N$ denotes the number of parameters. Our scaling law accurately captures the training loss across different training durations.

where $N_C$ represents the constraint on model size and $D_C$ denotes the constraint on the number of training tokens. To account for the inference latency budget, we introduce a new term $T_C$ to our scaling law formulation to represent the inference latency constraint.

Motivated by Figure 2, we closely examine the effect of the aspect ratio ($d_{\mathrm{model}}/n_{\mathrm{layers}}$) on inference latency and throughput by altering the hidden size $d_{\mathrm{model}}$ and the number of layers $n_{\mathrm{layers}}$ as shown in Figure 3 and Figure 4. Reasonable aspect ratios are chosen based on open-weight models listed in Appendix G.

Figure 3(a) shows that inference latency increases linearly with the number of layers when the hidden size remains constant. This occurs as the inference computation must be performed sequentially, one layer at a time (Yan et al., 2024). However, the matrix computations within a single layer can be performed in parallel. Furthermore, Figure 3(c) indicates that for the same number of parameters, we can

achieve different latency targets by changing the ratio of the number of hidden parameters in one layer ($d_{\mathrm{model}}$) vs. the number of layers ($n_{\mathrm{layers}}$). Moreover, in Figure 4, We study the relationship between model shape and inference throughput under a fixed parameter budget. We observe that, under a fixed parameter budget, wider and shallower models consistently achieve higher inference throughput. Due to space constraints, results on the relationship between aspect ratio and time to first token (TTFT) are provided in Appendix C.

Prior work (Kaplan et al., 2020) has shown the impact of the aspect ratio ($d_{\mathrm{model}}/n_{\mathrm{layers}}$) on the performance of the model. However, it does not define the connection between model size, number of training tokens, and model shape. To establish this relationship, we trained several small models $N \in \{80, 116, 164, 237, 313\}$M by varying the aspect ratio and setting $D \in \{20, 40, 160\}N$. Due to resource limitations, we only train a subset of the models at $D = 160N$.

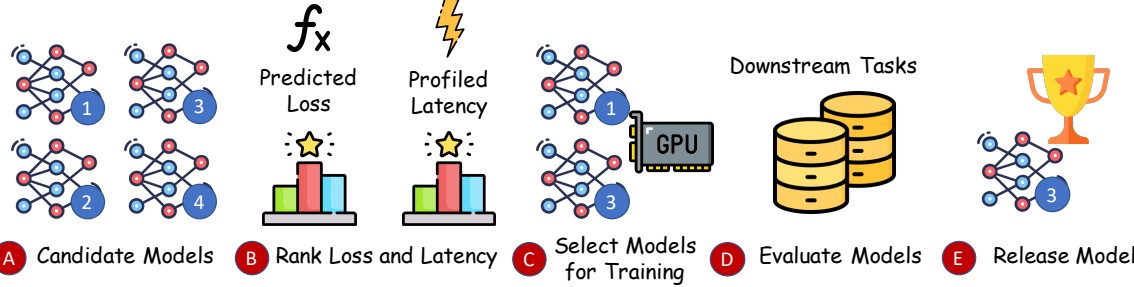

**A** Candidate Models  **B** Rank Loss and Latency  **C** Select Models for Training  **D** Evaluate Models  **E** Release Model

*Figure 6.* **An Overview of Methodology: (A)** The model training team first selects several candidate models with various model sizes and configurations; **(B)** Measure the inference latency using open-source inference systems and predict model loss with fitted scaling laws; **(C)** Select top-$k$ candidate models for training based on inference latency and loss; **(D)** Evaluate the models over downstream tasks after training; **(E)** Release the best model based on inference efficiency and performance over downstream tasks.

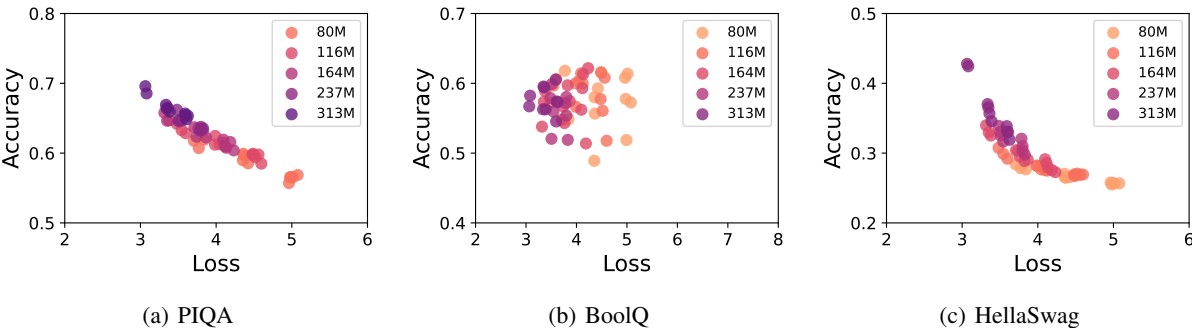

(a) PIQA  (b) BoolQ  (c) HellaSwag

*Figure 7.* **Accuracy vs. Loss:** (Left) We illustrate the correlation between accuracy and model loss on PIQA (Bisk et al., 2020). (Center) We present the connection between accuracy and model loss on BoolQ (Clark et al., 2019). (Right) We show the connection between accuracy and model loss on HellaSwag (Zellers et al., 2019). These three patterns shown in the plots demonstrate the difficulty in robustly predicting individual downstream task accuracies from scaling laws.

We plot the loss values against the aspect ratio in Figure 5. From the figure, we can see that the most suitable model shape adjustment is the inclusion of the term $(1 + \varepsilon R^\gamma)$ to the Chinchilla scaling law (Hoffmann et al., 2022). Therefore, we derive the following inference-efficient scaling law formulation:

$$L(N, D, R) = (E + AN^{-\alpha} + BD^{-\beta}) \cdot (1 + \varepsilon R^\gamma) \quad (4)$$

where $N$ is the number of parameters, $D$ is the number of training tokens, and $R = d_{\text{model}}/n_{\text{layers}}$ is the aspect ratio. Moreover, $A, B, E, \alpha, \beta, \gamma, \varepsilon$ are learned parameters. In Figure 5, we plot the predicted values from the scaling law against the observed values from training. More details of the experimental setup and fitting procedure can be found in §3.

### 2.3. Methodology

Scaling laws were first developed to predict the loss of language models. However, LLMs are evaluated on the *performance of downstream tasks*. A recent study (Gadre et al., 2024) attempts to establish scaling laws that link evaluation

loss to errors in downstream tasks. Inherently, predicting the error in downstream tasks becomes challenging when model losses are similar, due to noise and inaccuracies in scaling laws. We observe this in Figure 7. To tackle this challenge, we develop a new method for training inference-efficient models, as shown in Figure 6. Our key idea is that inference latency measurement has negligible overhead, and scaling laws can help us estimate the loss of scaled-up models. Thus, we propose identifying top-$k$ candidate models using inference latency and loss data, where the user can choose $k$. After training, we evaluate these models on downstream tasks and release the best-performing model to the public, taking into account both inference latency and performance on downstream tasks. Our method (Figure 6) can also be applied to different architectural optimizations, such as MLA (Liu et al., 2024a), to quantify the accuracy-efficiency tradeoff.

# 3. Experiments

We next discuss the experiment setup we use for model training and evaluation (§3.1). Following that, in §3.2, we demonstrate how to fit scaling laws using our experimental results.

## 3.1. Experimental Setup

**Training Setup.** For all experiments, we train transformer-based decoder-only language models (Vaswani, 2017). Following (Gururangan et al., 2023; Gadre et al., 2024), the model's architecture is similar to GPT-2 (Radford et al., 2019) and LLaMA (Touvron et al., 2023a), with GPT-NeoX (Black et al., 2022) employed as the tokenizer. We train models with a maximum of 1.5 billion parameters for up to 30 billion tokens, following the compute-optimal setup in (Hoffmann et al., 2022). The models are trained on uniformly sampled subsets of DCLM-Baseline (Li et al., 2024) with one epoch, ensuring no repetition in data (other than possible data repetition in the dataset itself). More details are included in Appendix A.

**Evaluation Setup.** We use HuggingFace (Wolf, 2019) to measure the inference efficiency of models over a single NVIDIA Ampere 40GB A100 GPU. By default, we set the number of input and output tokens to be 128 and 256, respectively, aligning with the distribution outlined in ShareGPT (Kwon et al., 2023).

We use LLM-foundry (llm, 2024) along with a zero-shot evaluation approach to evaluate model performance on downstream tasks. We evaluate the downstream task accuracy of models derived from the methodology outlined in §2.3 using the following datasets: ARC-Easy (Clark et al., 2018), ARC-Challenge (Clark et al., 2018), BoolQ (Clark et al., 2019), COPA (Roemmele et al., 2011), HellaSwag (Zellers et al., 2019), LAMBADA (Paperno et al., 2016), PIQA (Bisk et al., 2020), WinoGrande (Sakaguchi et al., 2021), MMLU (Hendrycks et al., 2020), Jeopardy (Jeo, 2022), and Winograd (Levesque et al., 2012).

Furthermore, to compare the predicted loss against the actual loss, we measure relative prediction error: $|\psi - \hat{\psi}|/\psi$, mean squared error (MSE): $\frac{1}{n}\sum_{i=1}^n (\psi_i - \hat{\psi}_i)^2$, and $R^2 = 1 - \sum_{i=1}^n (\psi_i - \hat{\psi}_i)^2 / \sum_{i=1}^n (\psi_i - \bar{\psi})^2$, where $\psi$ represents the actual loss, $\hat{\psi}$ the predicted loss from scaling laws, and $\bar{\psi} = \frac{1}{n}\sum_{i=1}^n \psi_i$. We also apply Spearman's rank correlation coefficient (Spearman, 1961) to evaluate how well the predicted rankings correspond to the actual rankings.

## 3.2. Fitting Scaling Laws

Following (Gadre et al., 2024), we use the Levenberg-Marquardt algorithm to fit Eq. (4). The Levenberg–Marquardt algorithm solves least-squares curve fitting problems, where the goal is to find the parameter vector $\beta$ of a model $f(x, \beta)$ that minimizes the sum of squared deviations. Formally, the problem can be expressed as $\arg\min_\beta \sum_{i=1}^m [y_i - f(x_i, \beta)]^2$, where $(x_i, y_i)$ are data pairs. Following observations from Chinchilla scaling law (Hoffmann et al., 2022) and another recent work (Gadre et al., 2024), we set $\alpha$, $\beta$, and $\gamma$ equal to simplify the fitting procedure. To fit and evaluate the scaling law, we train 63 models using a range of model sizes, shapes, and amounts of training tokens. The size of our model ranges from 80M to 339M and the number of tokens used for training ranges from 1.6B to 12.8B. Detailed model configurations can be found in Table 4 in Appendix A.

# 4. Results

In this section, we first study the predictive power of our inference-efficient scaling laws in §4.1. Then, in §4.2, we release an inference-efficient model that maintains accuracy on downstream tasks compared with open-sourced models by using the methodology outlined in Figure 6. We also show that our method significantly outperforms Chinchilla in predicting the best model configurations. Finally, we perform ablation studies on obtaining robust scaling laws and show that our inference-efficient scaling law is more robust than Chinchilla in various scenarios in §4.3.

*Table 1.* **Data Used to Fit Scaling Laws:** In this table, we show the number of parameters and tokens used in model training to fit the scaling laws in Figure 8-10. ✓ indicates we use all model variants with the given size and ✗ means we do not use any model variants with the given size. ❖ indicates that we randomly sample one model variant from the candidate set. The details of model variants are included in Appendix A.

| $N$ | $D$ | Figure 8 | Figure 9 | Figure 10 |
|------|------|----------|----------|-----------|
| 80M | 1.6B | ✓ | ✓ | ❖ |
| 116M | 2.3B | ✓ | ✓ | ❖ |
| 164M | 3.2B | ✓ | ✓ | ❖ |
| 237M | 4.7B | ✓ | ✓ | ❖ |
| 313M | 6.2B | ✓ | ✓ | ❖ |
| 80M | 12.8B | ✓ | ✗ | ❖ |

## 4.1. Prediction acccuracy

As shown in §3.2, we obtain the actual losses of various models by training multiple small models with different model configurations to establish the scaling law. We set $N \in \{80, 116, 164, 237, 313\}$M and $D = 20N$ to train small models and collect the data to fit the learnable parameters in Eq. (2) and Eq. (4). Furthermore, to enhance the generality of the scaling law, we train 80M models with $D = 160N$ tokens, thereby collecting data from an over-

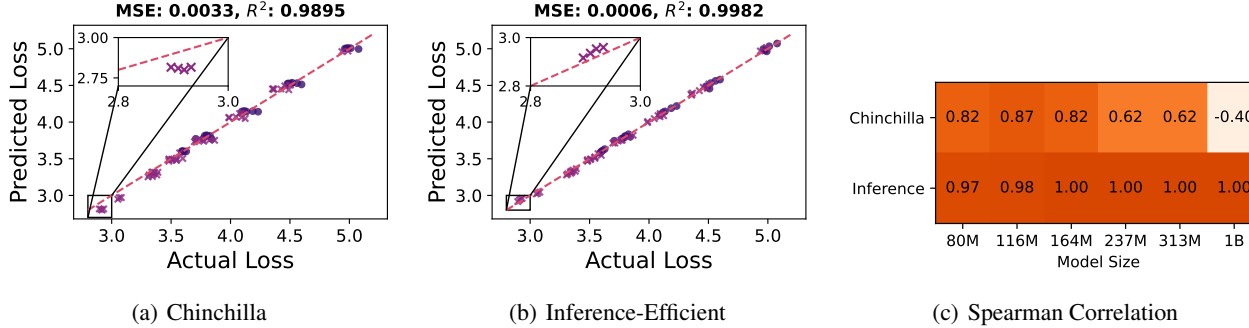

(a) Chinchilla      (b) Inference-Efficient      (c) Spearman Correlation

*Figure 8.* **Comparison:** (Left) We illustrate the predicted versus actual loss using Eq. (2). (Center) We display the comparison of predicted to actual loss based on Eq. (4). Dots represent data points used for curve-fitting, while cross marks represent test data points. (Right) We demonstrate that our inference-efficient scaling law yields a significantly higher Spearman correlation, resulting in more precise predictions of the optimal model configuration.

training setting.

Then, we train larger models on more tokens to evaluate the predictive power of our inference-efficient scaling law. We present the results in Figure 8. Figure 8 demonstrates that our scaling law achieves higher accuracy than the Chinchilla scaling law, as shown by a smaller MSE and a larger $R^2$ (Wright, 1921) value. We reduce MSE from 0.0033 to 0.0006 while improving $R^2$ from 0.9895 to 0.9982. In addition, the relative prediction error for the inference-efficient scaling law is less than $1.2\%$, whereas for the Chinchilla scaling laws, it ranges from $2.7\%$ to $4.1\%$. This demonstrates that the inference-efficient scaling law predicts more accurately than the Chinchilla scaling law.

Furthermore, as illustrated in Figure 6, prioritizing the ranking of predicted loss is more critical than its absolute value when employing the training methodology described in §2.3 for inference-efficient models. We calculate Spearman's rank correlation coefficient (Spearman, 1961) for both the Chinchilla scaling law and the inference-efficient scaling law when predicting the loss of 1B models. The results are shown in Figure 8(c). The results indicate that our inference-efficient law is more effective in ranking different model configurations. For example, the inference-efficient scaling law shows a Spearman correlation of 1.00 for the 1B model loss prediction, in contrast to Chinchilla's -0.40. In Appendix A, we include more details on model configurations.

### 4.2. Inference-Efficient Models

Guided by the accurate inference-efficient scaling law, we employ the predict, rank, and select method outlined in Figure 6 to train inference-efficient models. First, we generate a range of variants from the Open-LM-1B model (Gururangan et al., 2023) by adjusting the aspect ratio. Then, we measure the inference latency of model variants on a single A100 GPU. Next, we select 3 models based on the

*Table 2.* **Inference-Efficient Models:** In this table, we compare the results of Morph-1B variants against other open pretrained models of similar size. The evaluation of large language models such as Open-LM-1B (Gururangan et al., 2023), OPT-1.3B (Zhang et al., 2022), Pythia-1.3B (Biderman et al., 2023), Neox-1.3B (Black et al., 2022) and OPT-IML-1.3B (Iyer et al., 2022) is summarized from (Gururangan et al., 2023).

| Models | $d_{model}$ | $n_{layers}$ | Avg. | Latency (s) |
|---|---|---|---|---|
| Open-LM-1B | 2048 | 24 | 0.49 | 3.61 |
| OPT-1.3B | 2048 | 24 | 0.50 | 2.55 |
| Pythia-1.3B | 2048 | 22 | 0.49 | 3.28 |
| Neox-1.3B | 2048 | 24 | 0.49 | 3.99 |
| OPT-IML-1.3B | 2048 | 24 | 0.54 | 2.54 |
| Morph-1B-v1 | 2048 | 24 | 0.52 | 3.61 |
| Morph-1B-v2 | 2560 | 16 | 0.52 | 2.57 |
| Morph-1B | 3072 | 12 | 0.52 | 1.96 |

measured inference latency and predicted loss, and train candidate models with the same training dataset. Finally, we evaluate the trained models over 20 downstream tasks and we outline the results in Figure 1 and Table 2.

As a baseline, the architecture of Morph-1B-v1 is identical to that of Open-LM-1B. The superior performance of Morph-1B-v1 over Open-LM-1B can be attributed to the higher quality DCLM-Baseline dataset (dcl, 2024). Additionally, OPT-IML-1.3B outperforms Morph-1B-v1 since it undergoes pre-training on 6x more unique tokens (180B vs 30B) followed by a fine-tuning stage (Iyer et al., 2022). Next, we train Morph-1B and Morph-1B-v2 which are derived from Morph-1B-v1 by modifying the aspect ratio. We use the same 30B tokens to train Morph-1B, Morph-1B-v1, and Morph-1B-v2. As illustrated in Table 2, the inference latency for Morph-1B-v1 is $1.8\times$ lower compared to Morph-

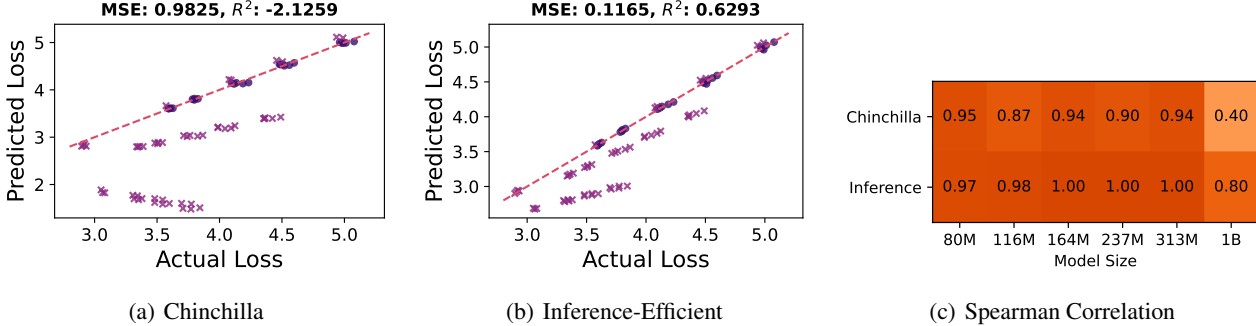

*Figure 9.* **Excluding Over-training Data:** We avoid using over-training data to fit the scaling laws. (Left) The figure is plotted by using Eq. (2). (Center) the center figure is created with Eq. (4). (Right) We plot the Spearman correlation of our scaling law versus the Chinchilla scaling law. The results indicate that additional training data can enhance the precision of scaling laws.

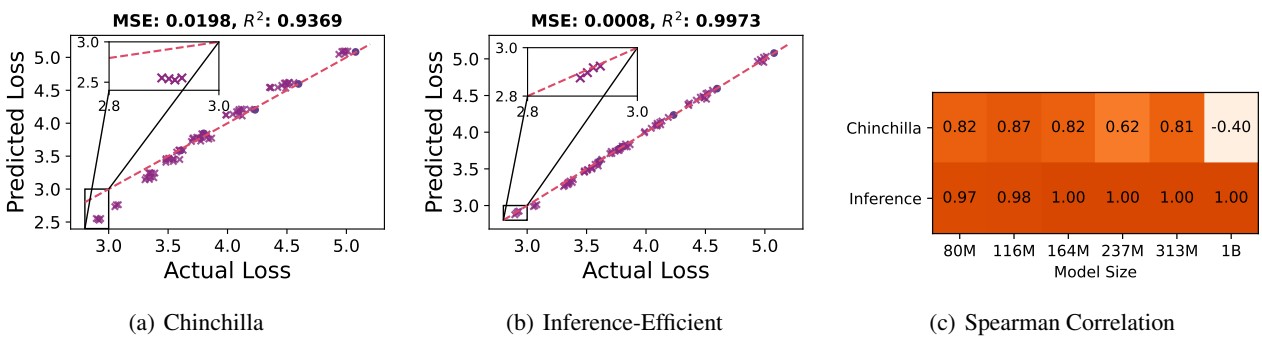

*Figure 10.* **Random Choice of Model Shape:** We randomly select the model shape to fit the scaling laws. (Left) The figure is plotted by using Eq. (2). (Center) The center figure is created with Eq. (4). (Right) We plot the Spearman correlation of our scaling law versus the Chinchilla scaling law. The results show that inference-efficient scaling laws are more robust than Chinchilla scaling laws.

1B, without any loss in accuracy.

### 4.3. Insights from Scaling Laws Fitting

Scaling laws provide a cheap and accurate way to predict language model performance at larger scales. However, a drawback of building scaling laws is the requirement to train models at various scales. In this section, we study how to make scaling laws robust and data-efficient.

**Exclude Over-training Data.** In this ablation study, we fit the scaling law based entirely on the Chinchilla-optimal setup, using only data points where training tokens are set to be Chinchilla-optimal. We vary the model size $N \in \{80, 116, 164, 237, 313\}$M and set the number of training tokens $D = 20N$, excluding data from $N = 80$M and $D = 160N$. Table 1 shows the configurations we run on and the results are shown in Figure 9. Compared to Figure 8, we observe that the inference-efficient scaling law is more robust than the Chinchilla scaling law. We achieve a much lower MSE of 0.1165 compared to Chinchilla's 0.9825 and an $R^2$ score of 0.6293 compared to Chinchilla's -2.1259.

However, we note that both scaling laws' performance deteriorates when applied to predicting losses in over-trained models. Therefore, data from over-training is essential to fit our inference-aware scaling law.

**Select Model Shape Randomly.** In this ablation study, we explore the robustness of our scaling laws via fitting models with random model architecture configurations. In this setting, the model architecture configuration for each size is chosen randomly. We randomly select a configuration from our model configuration pools (The complete list of candidate configurations can be found in Table 4 in Appendix). Figure 10 shows the experiment results. Compared to Chinchilla scaling laws, our inference-efficient scaling laws exhibit greater robustness with much smaller MSE (0.0008 vs 0.0198) and higher $R^2$ value (0.9973 vs 0.9369). We then use these two laws to predict the loss of 1B models. The results show that the relative prediction error for the inference-efficient scaling law is less than 0.72%, significantly lower than the Chinchilla scaling law's relative prediction error, which ranges from 11.8% to 13.4%. Finally, by using only six data points to fit the two scaling

laws, we significantly reduce the training costs associated with developing these laws. The GPU hours for fitting have been reduced from 450 to 85 A100 GPU hours.

## 5. Related Work

**Large Language Models.** Transformer (Vaswani, 2017) has been successfully applied to a variety of tasks: text classification (Wang, 2018; Sarlin et al., 2020), generation (Zellers et al., 2019; Sakaguchi et al., 2021), reasoning (Srivastava et al., 2022), and mathematics (Cobbe et al., 2021; Hendrycks et al., 2021), showcasing their broad applicability and effectiveness. The development of the GPT models (Brown et al., 2020) demonstrates that increasing the scale of language models significantly enhances their performance across various downstream tasks. The success of the GPT models has inspired the subsequent development of many large language models, including but not limited to LLaMA (Touvron et al., 2023a;b), Gemma (Team et al., 2024a;b), Qwen (Bai et al., 2023; Yang et al., 2024), and DeepSeek (Liu et al., 2024a;b; Guo et al., 2025), each designed to push the boundaries of language modeling.

**Scaling Laws.** Scaling laws are powerful predictors for how large language models behave as parameters increase (Kaplan et al., 2020). Plenty of subsequent works have contributed to the development of scaling laws (Hoffmann et al., 2022; Muennighoff et al., 2023; Sardana et al., 2023; Tao et al., 2024; Kumar et al., 2024; Gadre et al., 2024; Ruan et al., 2024; Abnar et al., 2025; Krajewski et al., 2024). In particular, Chinchilla scaling law (Hoffmann et al., 2022) optimizes a fixed computing budget allocation by balancing the number of model parameters against the number of training tokens to minimize the training loss. Data-Constrained scaling law (Muennighoff et al., 2023) extends the Chinchilla scaling laws by considering repeated data. The scaling laws presented in (Gadre et al., 2024) not only predict training loss under over-training scenarios but also connect training loss to downstream error. Beyond Chinchilla-Optimal (Sardana et al., 2023) attempted to account for inference cost in their scaling law. However, unlike training tokens, the number of inference tokens cannot be measured in advance.

**Inference Serving Systems.** Inference cost has drawn significant attention in recent years. Many inference systems and algorithms have been developed to speed up model serving (Olston et al., 2017; Gujarati et al., 2020; Gugger et al., 2022; Yu et al., 2022; Leviathan et al., 2023; Kwon et al., 2023; Zheng et al., 2023; Agarwal et al., 2024a;b; Ye et al., 2025; MLC team, 2023-2025). Specifically, Orca (Yu et al., 2022) utilizes continuous batching to achieve higher inference throughput. vLLM (Kwon et al., 2023) improves the throughput of popular LLMs by using PagedAttention to manage the KV cache memory. Furthermore, SGLang (Zheng et al., 2023) improves the inference throughput and latency by using RadixAttention. A recent study introduces FlashInfer (Ye et al., 2025), which employs block-sparse and composable formats to tackle KV cache storage heterogeneity.

**Compute-Efficient Model Design.** Previous research has explored the trade-offs of various model configurations in Vision Transformers (ViTs) (Alabdulmohsin et al., 2023). Additionally, (Tay et al., 2021) demonstrates that training deep and narrow models can be particularly beneficial when computational resources are limited. More recently, several efficient attention mechanisms (Xiao et al., 2023; Gao et al., 2024; Jiang et al., 2024; Xiao et al., 2024; Yuan et al., 2025) have been introduced to enhance inference efficiency by modifying the attention block.

## 6. Limitations and Future Work

Although there has been notable progress by our team, several unresolved challenges open up promising prospects for further study. First, due to resource limitations, we are unable to scale our training to include 7B models. Second, recently developed inference systems (Ye et al., 2025) can enhance inference efficiency and create new trade-offs between inference efficiency and model performance. Furthermore, Attention modules like Multi-Query Attention (MQA) (Shazeer, 2019), Grouped-Query Attention (GQA) (Ainslie et al., 2023) and Multi-Head Latent Attention (MLA) (Liu et al., 2024a) might also influence loss and inference latency. Our work provides a flexible way to quantify and predict how these architectural optimizations affect the accuracy-efficient tradeoffs. We hope this work opens up a new line of research that takes inference efficiency as an essential factor in designing language models.

## 7. Conclusion

In this work, we perform an extensive empirical study to develop scaling laws that guide us in designing inference-efficient model architecture. We first demonstrate that model architecture impacts inference efficiency and that existing scaling laws do not account for inference costs. To jointly optimize inference cost and model loss, we propose inference-efficient scaling laws. We conduct count number, each point is a number experiments to fit and evaluate the inference-efficient scaling laws. To tackle the disparity between model loss and downstream task performance, we have developed a novel methodology to train and rank inference-efficient models using our scaling law. Finally, we design and train Morph-1B model by leveraging inference-efficient scaling law, which enhances inference efficiency while maintaining accuracy in downstream tasks, compared to similar-sized open-sourced models.

# Acknowledgements

We gratefully acknowledge the support of the NSF Diamond project OAC-2311767 (Democratizing Large Neural Network Model Training for Science). We thank Zhao Zhang from Rutgers University for providing us access to computing resources and Vaishaal Shankar for assisting in using the DCLM dataset. The authors acknowledge the Texas Advanced Computing Center (TACC) at The University of Texas at Austin for providing computational resources that have contributed to the research results reported within this paper. This research also used computational resources from the NSF Cloudlab (Duplyakin et al., 2019) facility.

# Impact Statement

This paper presents work that aims to advance the field of Machine Learning. Our work aims to train more inference-efficient language models, potentially reducing the deployment cost of these models and their associated environmental impacts.

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

*Table 3.* **Hyperparameters:** We show the hyperparameters used for training in this paper. In addition, the batch size is the global batch size and the default sequence length is 2048.

| Model Size | Warmup | Learning rate | Weight decay | z-loss | Batch size |
|---|---|---|---|---|---|
| <400M | 2000 | 3e-3 | 0.033 | 1e-4 | 512 |
| 1B | 5000 | 3e-3 | 0.033 | 1e-4 | 256 |

*Table 4.* **Model Architectures:** We list the architectural configurations of all models trained in this paper. $d_{\text{model}}$ is the hidden size, $f_{\text{size}}$ is the intermediate size, $n_{\text{layers}}$ is the number of layers, and $n_{\text{heads}}$ is the number of attention heads.

| Model Size | Variant | $d_{\text{model}}$ | $f_{\text{size}}$ | $n_{\text{layers}}$ | $n_{\text{heads}}$ |
|---|---|---|---|---|---|
| 80M | v1 | 512 | 1536 | 8 | 8 |
| 80M | v2 | 576 | 1536 | 5 | 8 |
| 80M | v3 | 640 | 1792 | 3 | 8 |
| 80M | v4 | 448 | 1280 | 13 | 8 |
| 80M | v5 | 384 | 1024 | 22 | 8 |
| 86M | v1 | 576 | 1536 | 7 | 8 |
| 86M | v2 | 640 | 1792 | 4 | 8 |
| 116M | v1 | 640 | 1792 | 10 | 10 |
| 116M | v2 | 720 | 2048 | 6 | 10 |
| 116M | v3 | 800 | 2304 | 4 | 10 |
| 116M | v4 | 880 | 2560 | 3 | 10 |
| 116M | v5 | 560 | 1536 | 15 | 10 |
| 116M | v6 | 480 | 1280 | 24 | 10 |
| 126M | v1 | 720 | 2048 | 8 | 10 |
| 126M | v2 | 800 | 2304 | 5 | 10 |
| 164M | v1 | 768 | 2048 | 12 | 12 |
| 164M | v2 | 864 | 2304 | 8 | 12 |
| 164M | v3 | 960 | 2560 | 6 | 12 |
| 164M | v4 | 1056 | 2816 | 4 | 12 |
| 164M | v5 | 1152 | 3072 | 3 | 12 |
| 178M | v1 | 864 | 2304 | 10 | 12 |
| 178M | v2 | 960 | 2560 | 7 | 12 |
| 237M | v1 | 896 | 2560 | 14 | 14 |
| 237M | v2 | 1008 | 2816 | 10 | 14 |
| 237M | v3 | 1120 | 3072 | 8 | 14 |
| 237M | v4 | 1232 | 3328 | 6 | 14 |
| 313M | v1 | 1024 | 2816 | 16 | 16 |
| 313M | v2 | 1152 | 3072 | 12 | 16 |
| 313M | v3 | 1280 | 3584 | 9 | 16 |
| 313M | v4 | 1408 | 3840 | 7 | 16 |
| 339M | v1 | 1152 | 3072 | 14 | 16 |
| Morph-1B | v1 | 2048 | 5632 | 24 | 16 |
| Morph-1B | v2 | 2560 | 6912 | 16 | 16 |
| Morph-1B | / | 3072 | 8192 | 12 | 16 |

## A. Hyperparameters and Model Architectures

We follow the hyperparameters mentioned in (Li et al., 2024; Gadre et al., 2024) with the specific details presented in Table 3. A cooldown rate of 3e-5 is used in all experiments. All models are trained in bfloat16 precision using the AdamW optimizer. The number of parameters is computed using `sum(p.numel() for p in model.parameters())`. To examine

how model architecture influences loss metrics and inference performance, we vary the model configuraitons. Architectural details are provided in Table 4.

# B. Results over A30 GPUs

In this section, we first evaluate the inference efficiency of open-source large language models (LLMs), aiming to develop a robust scaling law across different hardware. From Figure 11, we observe similarly that both the number of parameters and the model architecture are crucial to the inference efficiency of the model. In addition, Figure 12 also demonstrates that inference latency increases linearly with the number of layers, and we can reduce inference latency by adjusting model configurations, which aligns with the observations made using the A100 GPU. Furthermore, we also evaluate the inference latency of models using various numbers of input and output tokens. Figure 13 demonstrates that the aforementioned conclusion remains valid when the number of input tokens is set to 1024 and the number of output tokens to 128.

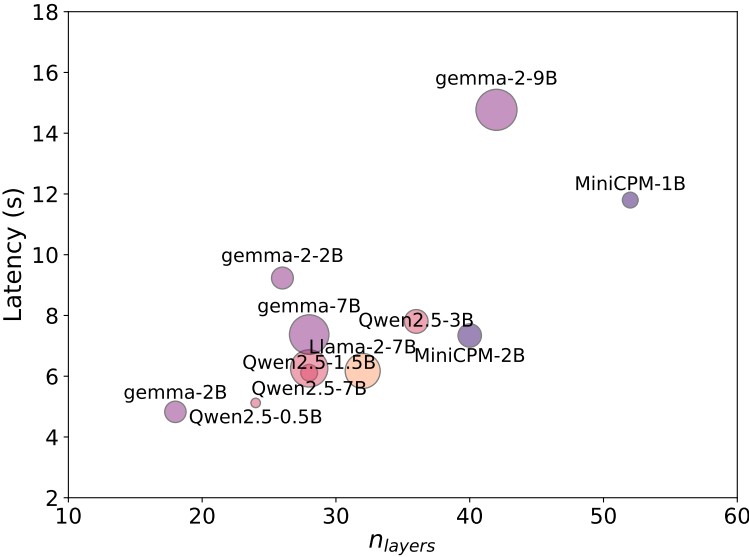

*Figure 11.* **Open-Source LLM's Inference Latency:** An overview of inference latency in open-source LLMs. The evaluated models include LLaMA (Touvron et al., 2023a), Qwen (Yang et al., 2024), Gemma (Team et al., 2024a;b), and MiniCPM (Hu et al., 2024). All evaluations were performed using the Hugging Face `generate` function on a single NVIDIA A30 Tensor Core GPU. In default, the number of input tokens is 128, and the number of output tokens is 256.

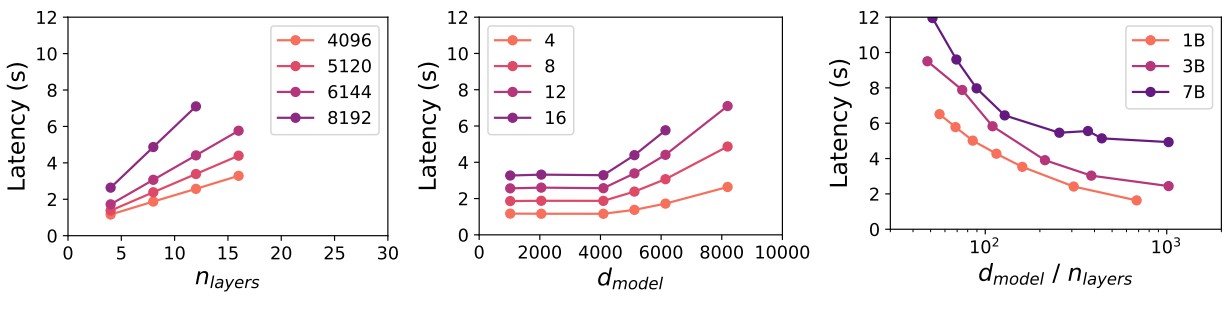

(a) Vary layers ($n_{\text{layers}}$), fix hidden size     (b) Vary hidden size ($d_{\text{model}}$), fix layers     (c) Vary ratio ($d_{\text{model}}/n_{\text{layers}}$), fix size $N$

*Figure 12.* **Model Shape on Inference Latency over A30 GPU:** (Left) We illustrate the correlation between inference latency and the number of layers, with the constant hidden size. (Center) We indicate the relationship between inference latency and hidden size with the number of layers fixed. (Right) We show the relationship between inference latency and aspect ratio, with the number of parameters fixed. All results are obtained using the Hugging Face `generate` function, with input and output token counts set at 128 and 256, respectively.

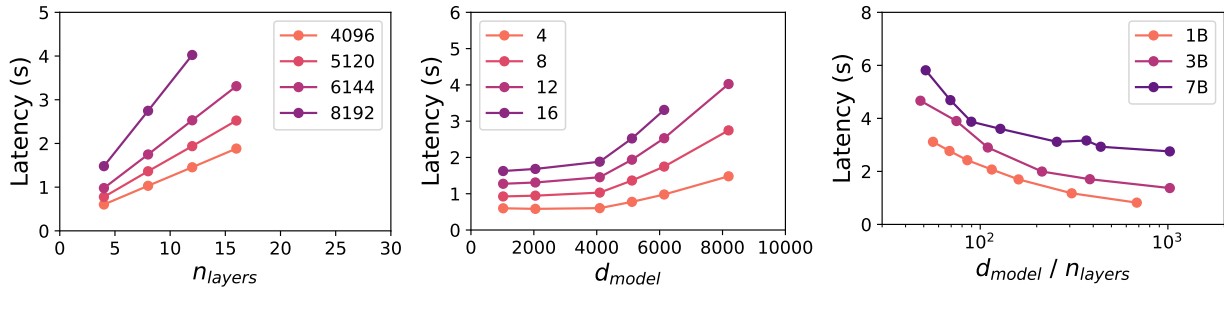

(a) Vary layers ($n_{\text{layers}}$), fix hidden size    (b) Vary hidden size ($d_{\text{model}}$), fix layers    (c) Vary ratio ($d_{\text{model}}/n_{\text{layers}}$), fix size $N$

*Figure 13.* **Model Shape on Inference Latency over A30 GPU with different number of input and output tokens:** (Left) We illustrate the correlation between inference latency and the number of layers, with the constant hidden size. (Center) We indicate the relationship between inference latency and hidden size with the number of layers fixed. (Right) We show the relationship between inference latency and aspect ratio, with the number of parameters fixed. All results are obtained using the Hugging Face `generate` function, with input and output token counts set at 1024 and 128, respectively.

## C. More Results over A100 GPUs

In this section, we evaluate the relationship between model architecture and Time To First Token (TTFT) over a single NVIDIA Ampere 40GB A100 GPU by fixing the total parameter count and varying the hidden size and number of layers. From Figure 14, we observe that wider and shallower models consistently achieve lower TTFT.

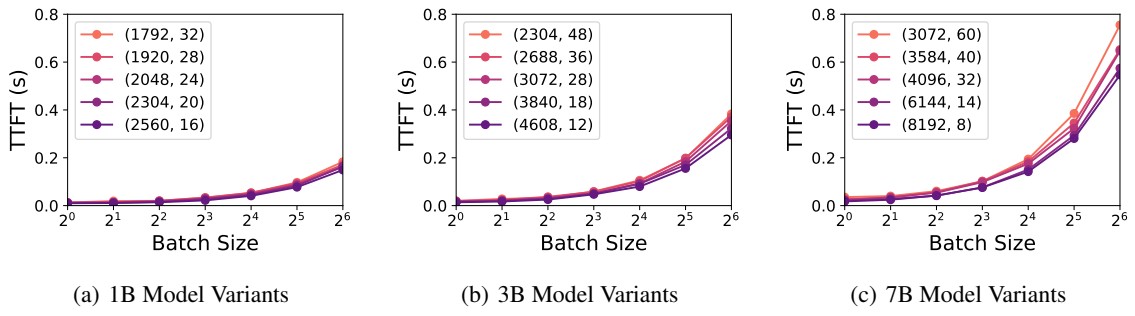

(a) 1B Model Variants      (b) 3B Model Variants      (c) 7B Model Variants

*Figure 14.* **Model Shape on Time To First Token (TTFT):** We examine the relationship between TTFT and model architecture by fixing the total parameter count and varying the hidden size and number of layers. Across different batch sizes, wider and shallower models consistently achieve lower TTFT. Each tuple in the legend represents a model configuration: the first number is the hidden size $d_{\text{model}}$, and the second is the number of layers $n_{\text{layers}}$. All evaluations were performed using the Hugging Face generate function on a single NVIDIA Ampere 40GB A100 GPU with input length 128, and output length 1.

# D. Results over vLLM

In this section, we first evaluate the inference efficiency of open-source large language models over vLLM using NVIDIA Tesla A100 Ampere 40 GB GPU. From Figure 15, we find that the efficiency of model inference is influenced not only by the number of parameters but also by the model's architecture. Additionally, Figure 16 shows that inference latency increases linearly with the number of layers using vLLM framework (Kwon et al., 2023). Modifying the model configurations effectively reduces inference latency, consistent with findings from the Hugging Face system (Wolf, 2019).

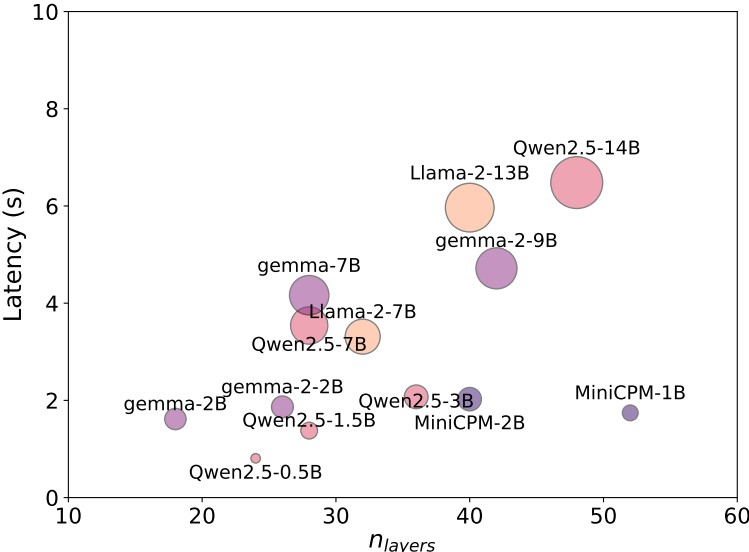

*Figure 15.* **Open-Source LLM's Inference Latency over vLLM (Kwon et al., 2023) using A100 GPU:** An overview of inference latency in open-source LLMs. The evaluated models include LLaMA (Touvron et al., 2023a), Qwen (Yang et al., 2024), Gemma (Team et al., 2024a;b), and MiniCPM (Hu et al., 2024). All evaluations were performed using the Hugging Face `generate` function on a single NVIDIA A100 Tensor Core GPU. In default, the number of input tokens is 128, and the number of output tokens is 256.

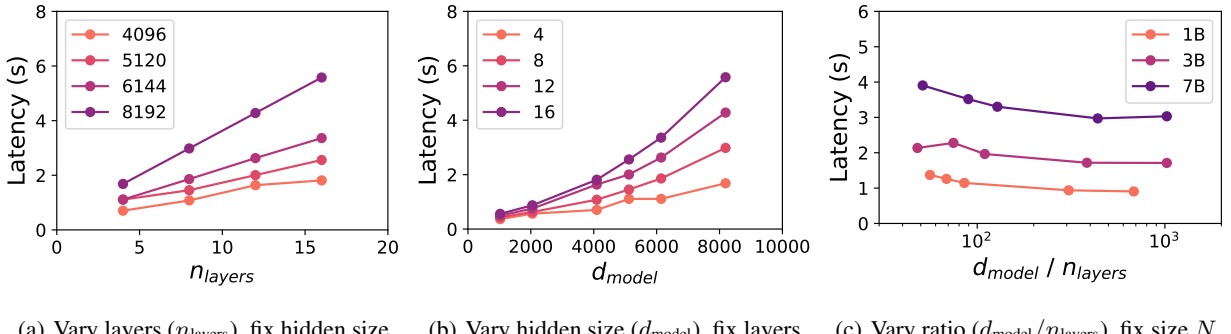

(a) Vary layers ($n_{\text{layers}}$), fix hidden size     (b) Vary hidden size ($d_{\text{model}}$), fix layers     (c) Vary ratio ($d_{\text{model}}/n_{\text{layers}}$), fix size $N$

*Figure 16.* **Model Shape on Inference Latency over vLLM (Kwon et al., 2023) using A100 GPU:** (Left) We illustrate the correlation between inference latency and the number of layers, with the constant hidden size. (Center) We indicate the relationship between inference latency and hidden size with the number of layers fixed. (Right) We show the relationship between inference latency and aspect ratio, with the number of parameters fixed. All results are obtained using the Hugging Face `generate` function, with input and output token counts set at 128 and 256, respectively.

# E. More Scaling Laws Fits

In Section 4.3, we explore how the random selection of model shapes affects Chinchilla scaling laws and inference-efficient scaling laws. We repeat the experiments three times and we show the remaining results in Figure 17 and Figure 18. We have similar observation from Figure 17 and Figure 18 that inference-efficient scaling laws are more robust than Chinchilla scaling laws.

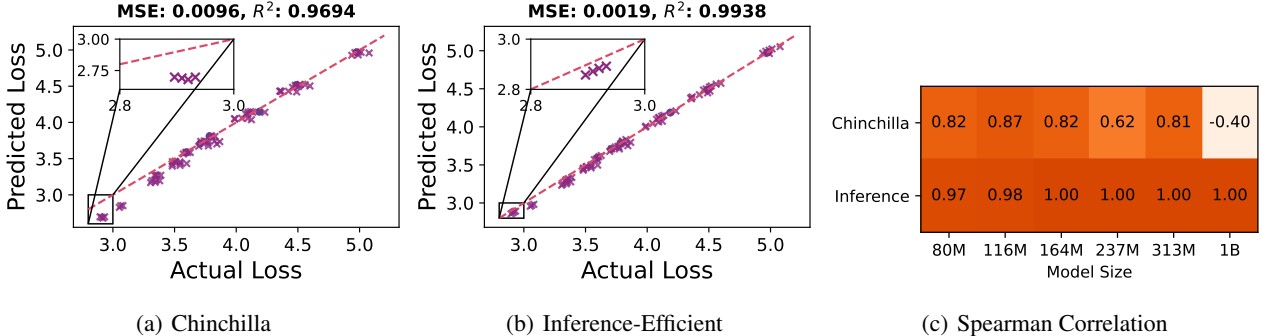

| (a) Chinchilla | (b) Inference-Efficient | (c) Spearman Correlation |

*Figure 17.* **Random Choice of Model Shape - Trial 2:** We randomly select the model shape to fit the scaling laws. (Left) The figure is plotted by using Eq. (2). (Center) The center figure is created with Eq. (4). (Right) We plot the Spearman correlation of our scaling law versus the Chinchilla scaling law. The models randomly selected from the fitting are 80M-v3-20N, 116M-v4-20N, 164M-v5-20N, 237M-v2-20N, 313M-v3-20N, and 80M-v4-160N.

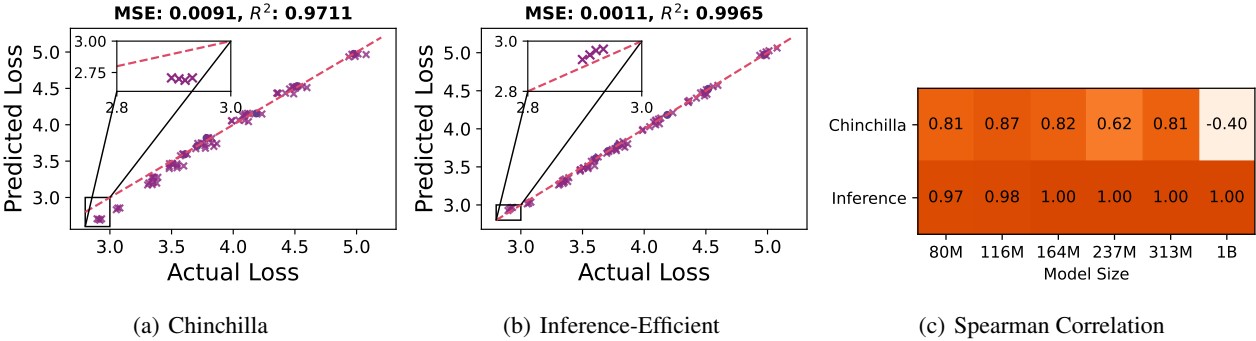

| (a) Chinchilla | (b) Inference-Efficient | (c) Spearman Correlation |

*Figure 18.* **Random Choice of Model Shape - Trial 3:** We randomly select the model shape to fit the scaling laws. (Left) The figure is plotted by using Eq. (2). (Center) The center figure is created with Eq. (4). (Right) We plot the Spearman correlation of our scaling law versus the Chinchilla scaling law. The models randomly selected from the fitting are 80M-v1-20N, 116M-v3-20N, 164M-v5-20N, 237M-v4-20N, 313M-v4-20N, and 80M-v4-160N.

# F. Evaluation Dataset Details

We include the details of the evaluation datasets in Table 5 and we use LLM-foundry (llm, 2024) to do all evaluations in this work.

Table 5. **Dataset Details:** We use LLM-foundry (llm, 2024) to do all evaluations.

| Dataset | Category | Evaluation Type |
| --- | --- | --- |
| ARC-Challenge (Clark et al., 2018) | world knowledge | multiple choice |
| ARC-Easy (Clark et al., 2018) | world knowledge | multiple choice |
| BoolQ (Clark et al., 2019) | reading comprehension | multiple choice |
| COPA (Roemmele et al., 2011) | commonsense reasoning | multiple choice |
| HellaSwag (Zellers et al., 2019) | language understanding | multiple choice |
| Jeopardy (Jeo, 2022) | world knowledge | language modeling |
| LAMBADA (Paperno et al., 2016) | language understanding | language modeling |
| MMLU (Hendrycks et al., 2020) | world knowledge | multiple choice |
| PIQA (Bisk et al., 2020) | commonsense reasoning | multiple choice |
| Winograd (Levesque et al., 2012) | language understanding | schema |
| WinoGrande (Sakaguchi et al., 2021) | language understanding | schema |

# G. Open-Source Model Configurations

In this section, Table 6 presents model configurations from Hugging Face, highlighting the vast space of architectural design choices.

Table 6. **Model Configurations:** We present the configurations of models available on Hugging Face.

| Model | $d_{\text{model}}$ | $n_{\text{layers}}$ | $d_{\text{model}} / n_{\text{layers}}$ |
| --- | --- | --- | --- |
| Llama-3.2-1B (Dubey et al., 2024) | 2048 | 16 | 128 |
| Llama-3.2-3B (Dubey et al., 2024) | 3072 | 28 | 109.7 |
| Qwen2.5-0.5B (Yang et al., 2024) | 896 | 24 | 37.3 |
| Qwen2.5-1.5B (Yang et al., 2024) | 1536 | 28 | 54.9 |
| Qwen2.5-3B (Yang et al., 2024) | 2048 | 36 | 56.9 |
| Qwen2.5-7B (Yang et al., 2024) | 3584 | 28 | 128 |
| Qwen2.5-14B (Yang et al., 2024) | 5120 | 48 | 106.7 |
| gemma-2b (Team et al., 2024a) | 2048 | 18 | 113.8 |
| gemma-7b (Team et al., 2024a) | 3072 | 28 | 109.7 |
| gemma-2-2b (Team et al., 2024b) | 2304 | 26 | 88.6 |
| gemma-2-9b (Team et al., 2024b) | 3584 | 42 | 85.3 |
| gemma-2-27b (Team et al., 2024b) | 4608 | 46 | 100.2 |
| microsoft-phi-2 (Phi, 2023) | 2560 | 32 | 80 |
| microsoft-phi-4 (Abdin et al., 2024) | 5120 | 40 | 128 |

# H. Parameter Fits

*Table 7.* **Parameter Fits:** Coefficients for the scaling laws presented in Figure 8.

| Law | A | B | E | $\alpha$ | $\epsilon$ |
|---|---|---|---|---|---|
| Chinchilla | 7720.62 | 68572.73 | 2.13 | 0.49 | / |
| Inference-Efficient | 54754.14 | 778340.38 | 2.45 | 0.61 | 0.0011 |

*Table 8.* **Parameter Fits:** Coefficients for the scaling laws presented in Figure 9.

| Law | A | B | E | $\alpha$ | $\epsilon$ |
|---|---|---|---|---|---|
| Chinchilla | -25287.67 | 248461.43 | 2.14 | 0.51 | / |
| Inference-Efficient | -16247.15 | 958437.97 | 2.41 | 0.60 | 0.0011 |

*Table 9.* **Parameter Fits:** Coefficients for the scaling laws presented in Figure 10.

| Law | A | B | E | $\alpha$ | $\epsilon$ |
|---|---|---|---|---|---|
| Chinchilla | 793.45 | 4090.85 | 1.09 | 0.35 | / |
| Inference-Efficient | 32515.16 | 408925.99 | 2.34 | 0.58 | 0.0016 |

# I. Contribution Statement

- Song collaborated with Minghao to set up the experimental environment and codebase and design experiments (§3). Song collected all experimental data and developed the inference-efficient scaling laws based on Chinchilla scaling laws (§2.2). Song proposed the methodology for training inference-efficient models (§2.3), conducted all experiments (§4), and prepared all figures in the paper. Ultimately, Song was responsible for writing and editing the paper.

- Minghao proposed investigating how model configuration affects LLM scaling in inference efficiency and performance. He collaborated with Song to set up the experimental environment and codebase and design experiments (§3), offered constructive suggestions throughout the project, and was responsible for writing and editing the paper.

- Shivaram provided assistance in shuffling DCLM datasets and recommended training a 1B model that is more inference-efficient while maintaining accuracy on downstream tasks compared with other models (§3). He provided numerous constructive comments throughout the project and helped us polish the entire paper.

