# OpenReview forum: "Scaling Inference-Efficient Language Models"
_ICML.cc/2025/Conference — ICML 2025 poster_

### Official Review · Reviewer_kVaw · 2025-03-01

**Overall Recommendation:** 2

**Summary:**

This paper proposes to modify the chinchilla scaling laws to also include the model aspect ratio (embedding dim / number of layers) into the scaling law.
This accounts for the fact that wider and shallower models are faster in inference.
Additionally the paper suggests to include the latency as key metrics for inference into the model selection criteria.
Finally, the authors train a wider and shallower 1B parameter model which has a lower latency than comparable models.

**Claims And Evidence:**

Claims:
> Models of similar training loss exhibit gaps in downstream evaluation.
> Also L. 88: “the disparity between model loss and accuracy in downstream tasks”.
- This statement contradicts findings in other papers ([1], [2]), which find that model loss is a good proxy for aggregate metrics on downstream tasks.
- This paper only looks at 3 downstream tasks and find that the have different correlations with the loss. For such general statements more downstream tasks should be considered and then very likely the observation as in [1] and [2] is confirmed.

> L.130 (Figure 2): smaller models can sometimes exhibit higher inference latencies than larger models
- Just the reference to Figure 2 is not enough. Why should this model be slower than a 14B model. I suspect problems with the Huggingface implementation. Needs more details and explanation

> The authors claim that their scaling law is inference aware.
- There is only an indirect relation to the inference efficiency: The authors find that wider, more shallow models have a smaller latency.

> The paper asks the question: Given dataset and parameter constraints, can we train an inference-efficient and accurate model for downstream tasks?
- In my view this question is too general. If I had to train an inference efficient model, I would look for more architecture interventions than just the aspect ratio, e.g. Linear Attention, Hybrids, MLA, etc.
- Either the paper has to demonstrate that their law (or modification to the law) holds for other attention variants, like Linear Attention, MLA or GQA or limit their scope.

[1] Sardana, Nikhil, et al. "Beyond chinchilla-optimal: Accounting for inference in language model scaling laws." arXiv preprint arXiv:2401.00448 (2023).

[2] Gadre, Samir Yitzhak, et al. "Language models scale reliably with over-training and on downstream tasks." arXiv preprint arXiv:2403.08540 (2024).

**Essential References Not Discussed:**

A recent work which studies the discrepancies between the Chinchilla Scaling law and the Hoffmann scaling law is relevant [4].


[4] Porian, T., Wortsman, M., Jitsev, J., Schmidt, L., & Carmon, Y. (2025). Resolving discrepancies in compute-optimal scaling of language models. Advances in Neural Information Processing Systems, 37, 100535-100570.

**Experimental Designs Or Analyses:**

- Details on the latency measurements are missing. Is torch.compile used? Are other optimizations conducted, e.g. torch CUDA graphs ?
- Details on the FLOP calculation used in this paper are missing. Are embedding FLOPs counted? What about last linear layer (i.e. unembedding) FLOPs?

- I did not find the final results of the fits in their paper, i.e. the values of the coefficients. It would be interesting how much their "chinchilla" fit deviates from the original "chinchilla" fit. This could help to assess how the papers experimental setup which contains models from 80M to 1B relates to the original "chinchilla" setup.

- The paper includes some ablations on the scaling law fits, where the data points for fitting the scaling law are selected differently.

- I would have expected additional ablations on different ways on accounting for the model aspect ratio, as this is the core contribution of this paper.

**Methods And Evaluation Criteria:**

The paper uses the MSE, R^2 value and the spearman correlation to compare the fits of the different scaling laws. I find these metrics suitable for comparing the curve fits.

However, I would expect some standard scaling law plots that show the loss over number of FLOPs or number of parameters.

L. 202: It is not obvious that the "most suitable model shape adjustment is the inclusion of the term $(1 + \epsilon R^\gamma)$. In my opinion this needs more discussion and motivation. Are there other possibilities to include the model shape?

**Other Comments Or Suggestions:**

- L. 292, Table 2: Caption: Typo "of of".

**Other Strengths And Weaknesses:**

Strengths:
- The paper studies the important question of how to build inference efficient language models.
- The paper studies the impact of the aspect ratio embedding dim / num layers on scaling laws.
- The paper tries to include key metrics for inference such as latency into scaling laws for language models.

Weaknesses:
- Only very small models are investigated.
  - It is not clear whether wider but more shallow models also perform that well in very large models (e.g. >7B parameters).
- The paper proposes to include the aspect ratio (embedding dim / number of layers) into the scaling law. However, from Figure 4 it seems that the loss is relative robust to the choice of this ratio, which suggests to not account for this.
  - I suggest to include a comment on other concurrent work, that suggests to fix this ratio (See Busbridge, et al., Distillation Scaling Laws, 2025 (http://arxiv.org/abs/2502.08606)
- Their methodology for selecting inference efficient language models seems to be motivated by the problem that predicting the downstream performance from language modeling loss is challenging.
  - However, instead of resolving this issue, the proposed method adds another selection step, where another metric (namely the latency) is used to select model candidates.
  - In my view the contribution of this proposed methodology is limited and not novel, as the "novel" method is to measure latency and select the fastest candidates, which is the setup of classical hyperparameter tuning.
- No other architectural changes than changing the aspect ratio between embedding dim and the number of layers are investigated.
  - For a paper that claims to scale “inference-efficient language models”, architectural changes such as GQA or MQA cannot be neglected.
  - At least there should be an attempt to investigate how these methods could be accounted for in the law.
  - Or it must be demonstrated that these methods have no impact on the scaling law.
- Just from Equation (4) it seems that only the aspect ratio is accounted for, but the aspect ratio alone does not affect the latency. Therefore, it is unclear how the constraint, that is formulated in equation (3) can be fulfilled.
- Their Morph-1B model is “just” a wider and shallower attention Transformer model.
  - It is not clear how the scaling law “guides” this model design (e.g. in number of parameters or number of training tokens).
  - It could have just been selected from the observation that wider and shallower models are faster on inference.

**Questions For Authors:**

- Why did the authors not include "standard" scaling law plots in there paper, where the loss is plotted over number of parameters or number of FLOPs?

**Relation To Broader Scientific Literature:**

In addition to the number of model parameters, the key contribution of the paper is a new scaling law formulation that incorporates the aspect ratio (embedding dim / num layers) of the model architecture into the scaling law. So while previous scaling law formulations (e.g. Chinchilla) depend on the number of parameters too, they remain unspecific about the allocation of the parameters in the model.

Concurrent works fix this ratio for scaling law experiments [3].

[3] Busbridge, Dan, et al. "Distillation Scaling Laws." arXiv preprint arXiv:2502.08606 (2025).

**Theoretical Claims:**

N/A

---

> ### Author Rebuttal · Authors · 2025-03-31
>
> **Claims**
>
> **C1:** This paper only looks...
>
> **Answer:** From Figure 3 in [2], prior work has also observed that models with smaller loss do not mean better performance over downstream tasks. See more results here: https://anonymous.4open.science/r/ICML25-Rebuttal-3B34
>
> **C2:** Just the reference to....
>
> **Answer:** Thanks for your question about the inference system and how it affects latency.  It is true that changing the inference system can change the latencies and we show results from vLLM in Figure 13, Figure 14. However, even then we see some interesting behavior where latencies for say the MiniCPM-1B is 30-40% higher than Qwen2.5-1.5B. We will clarify this point and include an explanation in the final version.
>
> **Experimental Design:**
>
> **E1:** Details on the latency...
>
> **Answer:** We used two inference frameworks in our work: hugging face (Figures 1, 2, 3, 10, 11, and 12) and vLLM (Figures 13 and 14). When using Huggingface we directly called the generate function. When using vLLM, we used torch.compile and torch CUDA graphs.
>
> **E2:** Details on the FLOP calculation...
>
> **Answer:** In this paper, we do not calculate FLOPs; instead, we directly measure the inference latency of models on GPUs. While FLOPs are a valuable metric, they do not accurately represent the real-world latency of a model [3]. Therefore, we focus on measuring the end-to-end inference latency.
>
> **Q1:** Only very small models are investigated....
>
> **A1:** Due to limited computational resources, we cannot afford to train larger models, such as a 7B model with over 140B tokens. We plan to expand our experiments to include larger models (>7B parameters) when we obtain more computational resources.
>
> **Q2:** The paper proposes to include the aspect...
>
> **A2:** We note that Distillation Scaling Laws was submitted to arXiv on Feb 12th, 12 days after the ICML deadline. We will discuss it in the final version.
>
> As for the aspect ratio, Figure 4 illustrates that with a fixed number of parameters, the loss can increase by up to 5% by varying the aspect ratio. In general, models with higher losses are more likely to exhibit poorer performance on downstream tasks, particularly when the loss increases by 5% or more. In addition, as shown in Figure 3, the model shape plays a crucial role in inference latency. Therefore, we believe we should account for this in the scaling law to co-optimize model loss and inference latency.
>
> **Q3:** Their methodology for selecting...
>
> **A3:** Thank you for the question. We would first like to clarify our pipeline here. In our pipeline, we incorporate inference latency into scaling laws, enabling them to accurately predict the loss ranking of candidate models in comparison to the Chinchilla law. However, due to the gap between loss and accuracy, as illustrated in Figure 5, we believe the effective solution is to evaluate models based on their performance over downstream tasks. Consequently, we choose the top-k models predicted by our scaling laws, pre-train them, and then evaluate their performance on downstream tasks. We then select the best model to release based on inference latency and downstream task performance. We note that both the updated formulation of our scaling law and how to integrate the proposed scaling law into the model selection pipeline are contributions of our work.
>
> **C3 & Q4:** No other architectural changes...
>
> **A4:** We agree with your points. In Section 6, “Limitations and Future Work,” we regard architectural changes such as GQA or MQA as future work.
>
> **Q5:** Just from Equation (4)...
>
> **A5:** As shown in Figure 3, with a fixed number of parameters, the aspect ratio predominantly affects the latency. Therefore, we want to solve the following question in the paper:
>
> $\arg\min L(N, D, R) = (E + AN^{−α} + BD^{−β} ) · (1 + εR^{γ})$
>
> s.t. $N \leq N_{C}$, $D \leq D_{C}$, $T_{inf} \leq T_{C}$
>
> **Q6:** Their Morph-1B model...
>
> **A6:** Given that we have a 24-layer base model Morph-1B-v1, we first propose several candidate models with lower inference latency. The configurations we used are (2560, 16), (3072, 12), (3200, 10), (3328, 8), (4096, 6), and (4608, 4), where the first element is the hidden size and the second element is the number of layers. Next, we use the scaling law to predict the loss of each model. Since our scaling laws can predict the rank of model loss more accurately than Chinchilla's, we selected the top 2 candidates (3072, 12) and (2560, 16) to do pre-training. After we obtain the pre-trained models, we evaluate them under downstream tasks and release the best one.
>
> **Q7:** Why did the authors not include "standard" scaling law plots...
>
> **A7:** https://anonymous.4open.science/r/ICML25-Rebuttal-3B34
>
> [1] Beyond chinchilla-optimal: Accounting for inference in language model scaling laws. arXiv 2023.
>
> [2] Language models scale reliably with over-training and on downstream tasks. arXiv 2024.
>
> [3] Run, don't walk: chasing higher FLOPS for faster neural networks. CVPR 2023.

---

> > ### Comment · Reviewer_kVaw · 2025-04-02
> >
> > I thank the authors for their effort in this answer.
> >
> > I see now there are now flop controlled experiments (isoFLOP curves) but instead the token param ratio is varied {20,40,160} this does not require FLOP calculation.
> > (Even though for the FLOP figures provided in the rebuttal, they must have been calculated somehow.)
> >
> > Note on the additional Figures: thanks the effort to create the additional figures. Would have been nice to mark the different models with different colors (e.g. different tokan param ratios or different model sizes in the loss-FLOP plot).
> >
> > In general my questions regarding Figure 4 remain. In the paper it is stated:
> > >We plot the loss values against the aspect ratio in Figure 4. From the figure, we can see that the most suitable model shape adjustment is the inclusion of the term $(1 + \epsilon R^\gamma)$
> >
> > Why does the term has this shape? I see that the dashed line (which is the proposed law) predicts the points reasonably well. How would the fit look without the term?
> > As you point to Figure 3: It is not clear to me how Latency (s) is measured (same as for Figure 1?). Typically inference performance in terms of speed is measured in Time to first Token (s) and Generation Throughput (toks/s) (or per token-latency).
> > E.g. this is a nice recent source on this: [https://jax-ml.github.io/scaling-book/inference/](https://jax-ml.github.io/scaling-book/inference/)
> >
> > Unfortunately, I am still not fully convinced so I tend to keep my score.

---

> > > ### Author Response · Authors · 2025-04-04
> > >
> > > We appreciate your response! Our answers are as follows:
> > >
> > > 1. We calculate FLOPs as 6ND, following the method outlined in [1], where N represents the number of parameters and D the number of tokens used in training.
> > > 2. We have updated the loss-FLOPs figures based on your recommendation of using different colors at https://anonymous.4open.science/r/ICML25-Rebuttal-3B34/FLOPs_rebuttal/loss_vs_flops_rebuttal.pdf
> > > 3. As for the shape of the term, similar to prior work [2] on scaling laws, our work is guided by the trend of losses observed across various model variants. The scaling laws without our new term will be the same as the Chinchiall scaling laws [1]. In Figures 7-9, we compare our scaling laws with the Chinchilla scaling law (i.e., with our term vs. without the term). We observe that our scaling laws significantly reduce the prediction error.
> > > 4. In terms of latency measurements, for Figure 3, we use the same setting as Figure 1 to measure the latency of models. All evaluations were performed using the Hugging Face generate function on a single NVIDIA Ampere 40GB A100 GPU with batch size 1, input length 128, and output length 256. We will clarify this in the paper.
> > > 5. We have more results here:
> > >
> > > To measure throughput (tokens / s), we fix the number of input tokens as 128 and the number of output tokens as 256 in all throughput experiments. All evaluations were performed using the Hugging Face generate function on a single NVIDIA Ampere 40GB A100 GPU
> > >
> > > To begin with, we fix the number of parameters and vary the batch size. The results of the 1B, 3B, and 7B models are shown:
> > > https://anonymous.4open.science/r/ICML25-Rebuttal-3B34/Figure3_rebuttal/tput/tput_vs_batch_size_1B_variants.pdf,
> > > https://anonymous.4open.science/r/ICML25-Rebuttal-3B34/Figure3_rebuttal/tput/tput_vs_batch_size_3B_variants.pdf,
> > > https://anonymous.4open.science/r/ICML25-Rebuttal-3B34/Figure3_rebuttal/tput/tput_vs_batch_size_7B_variants.pdf.
> > > The legend displays the model's hidden size / n_layers ratio. **We find that models with wider dimensions achieve higher throughput than narrower models with the same parameter count.**
> > >
> > > Then, we fix the hidden size and vary the batch size. The results are shown here:
> > > https://anonymous.4open.science/r/ICML25-Rebuttal-3B34/Figure3_rebuttal/tput/tput_vs_batch_size_with_layers_hidden_size_4096.pdf.
> > > The legend shows the number of layers of the model. We observe that throughput decreases linearly with the number of layers when the hidden size remains constant. We also have similar observations in https://anonymous.4open.science/r/ICML25-Rebuttal-3B34/Figure3_rebuttal/tput/tput_vs_batch_size_with_hidden_size_layer_4&8.pdf. In the legend, "4+4096" represents a model with 4 layers and a hidden size of 4096, while "8+8192" denotes a model with 8 layers and a hidden size of 8192.
> > >
> > > Finally, we fix the number of layers to 8 and vary the batch size. The results are shown here:
> > > https://anonymous.4open.science/r/ICML25-Rebuttal-3B34/Figure3_rebuttal/tput/tput_vs_batch_size_with_hidden_size_layer_8.pdf. We've noticed that models with a consistent number of layers but a smaller hidden dimension consistently achieve higher throughput.
> > >
> > > To measure Time To First Token (TTFT), we fix the number of input tokens as 128 and the number of output tokens as 1 in all TTFT experiments. All evaluations were performed using the Hugging Face generate function on a single NVIDIA Ampere 40GB A100 GPU.
> > >
> > > We first fix the number of parameters and vary the batch size. The results of the 1B, 3B, and 7B models are shown:
> > > https://anonymous.4open.science/r/ICML25-Rebuttal-3B34/Figure3_rebuttal/ttft/ttft_vs_batch_size_1B_variants.pdf,
> > > https://anonymous.4open.science/r/ICML25-Rebuttal-3B34/Figure3_rebuttal/ttft/ttft_vs_batch_size_3B_variants.pdf,
> > > https://anonymous.4open.science/r/ICML25-Rebuttal-3B34/Figure3_rebuttal/ttft/ttft_vs_batch_size_7B_variants.pdf.
> > > The legend indicates the ratio of hidden size to n_layers in the model. **We observe that models with a wider configuration exhibit lower TTFT compared to narrower models, given the same number of parameters.**
> > >
> > > Additionally, we illustrate the correlation between TTFT, layers, and hidden size in the PDF files provided, following a similar experimental setup as our throughput tests.
> > > https://anonymous.4open.science/r/ICML25-Rebuttal-3B34/Figure3_rebuttal/ttft/ttft_vs_batch_size_with_hidden_size_layer_4&8.pdf,
> > > https://anonymous.4open.science/r/ICML25-Rebuttal-3B34/Figure3_rebuttal/ttft/ttft_vs_batch_size_with_hidden_size_layer_8.pdf, and https://anonymous.4open.science/r/ICML25-Rebuttal-3B34/Figure3_rebuttal/ttft/ttft_vs_batch_size_with_layers_hidden_size_4096.pdf.
> > >
> > > [1] Hoffmann, Jordan, et al. "Training compute-optimal large language models." arXiv preprint arXiv:2203.15556 (2022).
> > >
> > > [2] Gadre, Samir Yitzhak, et al. "Language models scale reliably with over-training and on downstream tasks." arXiv preprint arXiv:2403.08540 (2024).

---

### Official Review · Reviewer_4YJ4 · 2025-03-09

**Overall Recommendation:** 5

**Summary:**

The paper observes that architecture modifications significantly affect inference latency whilst holding total model size fixed. The paper primarily uses model aspect ratio $r=d_{model}/n_{layers}$ as the parameterization of architecture.

The paper then introduces an inference-efficient scaling law, which is sensitive to $r$ and enables constraint optimization subject to a size, data and latency constraint.

The resulting process enables systematic improvement of the efficiency-accuracy pareto frontier.

## Update after rebuttal

I am maintaining my score of 5: Strong Accept. I can see this significantly deviates from some reviews presented by other reviewers.

The reasons I think this paper is well worth accepting are:
1. The empirical procedure is sound
2. The results are of practical value, although they are i) hardware and ii) architecture/task-constrained

The limitation of point 2i) is necessary in order to show a practical use case. A theoretical estimate of latency could also work here, but in general it's not clear how to map this estimate to scenarios of interest.

The task-constrained limitation is a consequence of essentially all empirical scaling law studies, and the authors should not be penalized for this.

The outstanding issue on the paper raised by another reviewer is why the specific form of the scaling law works from a first principles perspective. It would be great to have clarification around it, however, for practical utility all that is required for empirical studies is correct boundary conditions, asymptotic behavior, and good empirical fit, each of which the results meet.

**Claims And Evidence:**

All primary claims in the paper are supported with clear, convincing explanations and evidence:
1. Aspect ratio $r$ influences inference latency
2. The inference-efficient scaling law (Equation 4) that incorporates aspect ratio is a good predictor of model cross-entropy
3. Following their constrained optimization procedure pushes the inference-accuracy pareto frontier.

There is a secondary claim which is supported by evidence in the paper, but I feel more further evaluation is required: "The inference-efficient scaling law is more robust than the Chinchilla scaling law" (see e.g. line 358).

First, "robust" is an overloaded term. It's true that the inference-efficient scaling law fits better than the Chinchilla law on the data provided (e.g. Fig 8c and 9c). However, it's unclear to me if this means the law is robust, which usually means with respect to some perturbation? This claim needs to be quantified/made explicit, e.g. using a bootstrap approach.

If instead the paper means to convey the law is more accurate, the data from the Chinchilla study is available, see [1]. As a test of the inference efficient scaling law, it would be useful to understand if it is a better fit on the Chinchilla data than the Chinchilla law (the aspect ratios of the models in Chinchilla are provided in the Chinchilla Appendix [2]). Such an observation/confirmation would be extremely useful to the community, beyond the context of targeting inference latency.

[1] Chinchilla Scaling: A replication attempt https://arxiv.org/abs/2404.10102

[2] Training Compute-Optimal Large Language Models https://arxiv.org/abs/2203.15556

**Essential References Not Discussed:**

No essential references are missing.

**Experimental Designs Or Analyses:**

I checked the soundness and validity of all experimental designs. The paper makes sensible choices throughout.

However I will make comments about two aspects of the investigation.

### Excluding over-training data

In Figure 8 the paper shows the scaling law fits and extrapolations for training with only Chinchilla-optimal models $D=20N$. From the form of the scaling law
$$
L(N,D)=E+AN^{-\alpha}+BD^{-\beta}
$$
when we use Chinchilla optimal models, then
$L(N)=E+AN^{-\alpha}+B(20N)^{-\beta}=E+AN^{-\alpha}+B^\prime(N)^{-\beta}$
Further, as you then set $\alpha=\beta$ (line 248), the equation becomes
$L(N)=E+(A+B^\prime)N^{-\alpha}=E+A^{\prime\prime}N^{-\alpha}$.
Even though $A^{\prime\prime}$ can be identified through your fitting procedure, $A$ and $B^\prime$ cannot (infinitely many solutions of numbers summing to $A^{\prime\prime}$), and so the independent effects of $N$ and $D$ are lost. I.e. the data does not enable identifying of scaling coefficients (which is what you are finding in practice).

To circumvent this issue cleanly with a minimal compute budget, IsoFLOP protocols are often used (e.g. [2]).

[2] Training Compute-Optimal Large Language Models https://arxiv.org/abs/2203.15556

### Choice of matched coefficients

In line 249 the authors set $\alpha=\beta=\gamma$. There is prior evidence $\alpha=\beta$, e.g. [2]. It is unclear to be why the choice to also set $\gamma$ to these values was made.

[2] Training Compute-Optimal Large Language Models https://arxiv.org/abs/2203.15556

**Methods And Evaluation Criteria:**

Yes, the datasets and benchmarks chosen are all sensible and relevant.

**Other Comments Or Suggestions:**

1. In line 437, you say that your approach "avoids estimating tokens generated". I would rephrase this, as it implies your approach is circumventing a problem in earlier work. To me it seems more like there is only a philosophical difference. The current work is concerned with model latency, which is an instantaneous measure, and useful to control. Prior work [3] is concerned with the total carbon footprint of a model during its lifetime. These constraints are both important.
2. I would not call Equation 4 an "inference-efficient" scaling law. To me, it is a scaling law that can estimate cross-entropy, and can take into account the model aspect ratio. In this sense, I would say Equation 4 is more like an "aspect-ratio sensitive/aware" law. Equation 4 can be used to then induce inference efficiencies, but that follows from downstream analysis, and not from Equation 4 itself.
3. Typo: there is a power of 2 missing in the MSE definition on line 271.
4. Provide your coefficient fits in the paper, preferably with bootstrap CIs.

[3] Beyond Chinchilla-Optimal: Accounting for Inference in Language Model Scaling Laws https://arxiv.org/abs/2401.00448

**Other Strengths And Weaknesses:**

The paper is extremely easy to read, is well-structured, and has well-made figures.

**Questions For Authors:**

1. What happens if you allow $\gamma\neq \alpha$?
2. Is there a proxy/notion of latency we can use which is hardware agnostic?
3. Do you observe any systematic relationship between model aspect ratio and downstream evaluation? (e.g. in [6] it is shown that denser models perform better on reasoning tasks at a fixed number of parameters that sparse MoEs).

[6] Mixture of Parrots: Experts improve memorization more than reasoning https://arxiv.org/abs/2410.19034

**Relation To Broader Scientific Literature:**

The work generalizes the constrained optimization training procedures of [2, 3] from compute optimal, to lifecycle compute optimal, to latency optimal within practical constraints.

The work is complementary to other investigations in architecture modification that can improve latency at a given model capability, for example Sparse MoEs [4, 5].

[2] Training Compute-Optimal Large Language Models https://arxiv.org/abs/2203.15556

[3] Beyond Chinchilla-Optimal: Accounting for Inference in Language Model Scaling Laws https://arxiv.org/abs/2401.00448

[4] Scaling Laws for Fine-Grained Mixture of Experts https://arxiv.org/abs/2402.07871

[5] Parameters vs FLOPs: Scaling Laws for Optimal Sparsity for Mixture-of-Experts Language Models https://arxiv.org/abs/2501.12370

**Theoretical Claims:**

No theoretical claims are made in the work.

---

> ### Author Rebuttal · Authors · 2025-03-31
>
> **Q1:** In line 437, you say that your approach "avoids estimating tokens generated". I would rephrase this, as it implies your approach is circumventing a problem in earlier work. To me it seems more like there is only a philosophical difference. The current work is concerned with model latency, which is an instantaneous measure, and useful to control. Prior work [3] is concerned with the total carbon footprint of a model during its lifetime. These constraints are both important.
>
> **A1:** Thanks for pointing it out! We agree that the total carbon footprint of a model during its lifetime is also important. We will rephrase the sentence in the final version.
>
> **Q2:** I would not call Equation 4 an "inference-efficient" scaling law. To me, it is a scaling law that can estimate cross-entropy and can take into account the model aspect ratio. In this sense, I would say Equation 4 is more like an "aspect-ratio sensitive/aware" law. Equation 4 can be used to then induce inference efficiencies, but that follows from downstream analysis and not from Equation 4 itself.
>
> **A2:** Thank you for the feedback. We will improve the name in the final version.
>
> **Q3:** Typo: there is a power of 2 missing in the MSE definition on line 271.
>
> **A3:** We will fix this typo in the final version.
>
> **Q4:** Provide your coefficient fits in the paper, preferably with bootstrap CIs.
>
> **A4:** For Figure 7, https://anonymous.4open.science/r/ICML25-Rebuttal-3B34/Figure_7_coefficient.png; For Figure 8, https://anonymous.4open.science/r/ICML25-Rebuttal-3B34/Figure_8_coefficient.png; For Figure 9, https://anonymous.4open.science/r/ICML25-Rebuttal-3B34/Figure_9_coefficient.png.
>
> **Q5:** What happens if you allow $\gamma \neq \alpha$
>
> **A5:** Thank you for your suggestion! Given that \gamma is the exponent of R, we often face overflow issues when utilizing scipy.optimize.curve_fit. To circumvent this problem, we can use the bounds parameter within the function. When removing the assumption that $\gamma = \alpha$ and applying necessary bounds to avoid overflow, we found that our scaling laws become more accurate compared to existing ones. In Figure 7, the R² will become 0.9985 (from 0.9982), and the MSE will become 0.0005 (from 0.0006). We will update the results in the final version.
>
> **Q6:** Is there a proxy/notion of latency we can use which is hardware agnostic?
>
> **A6:** In this study, we advocate for designing a wider and shallower Transformer model. This approach is based on the understanding that inference in models occurs sequentially, layer by layer, while GPUs can process each layer concurrently. Therefore, a wider and shallower transformer model has smaller inference latency is a hardware-agnostic notion, though the degree would differ based on the specific hardware.
>
> **Q7:** Do you observe any systematic relationship between model aspect ratio and downstream evaluation? (e.g. in [6] it is shown that denser models perform better on reasoning tasks at a fixed number of parameters that sparse MoEs).
>
> **A7:** Thanks for your question! However, we do not observe any systematic relationship between the model aspect ratio and downstream evaluation. We will include this in future work.
>
> [3] Beyond Chinchilla-Optimal: Accounting for Inference in Language Model Scaling Laws https://arxiv.org/abs/2401.00448
>
> [6] Mixture of Parrots: Experts improve memorization more than reasoning https://arxiv.org/abs/2410.19034

---

> > ### Comment · Reviewer_4YJ4 · 2025-04-03
> >
> > I thank the authors for addressing my questions, as well as those of other reviewers.
> >
> > I have also considered the points raised by other reviewers. The only problematic one for me is raised by kVaw regarding why exactly the aspect ratio should be represented as $(1+\epsilon R^\gamma)$ and do agree with kVaw that a reader would benefit from following the thought process leading to this conclusion.
> >
> > Overall this does not change my perspective on whether the contribution is:
> >
> > i) Correct (it is, to the best of my assessment), or
> >
> > ii) Useful. I do agree with reviewer B9RC that our confidence in extrapolation is partially limited by the range of model sizes in the "test set" for this experiment (which are not too large). Without significant compute resources, this is true of all scaling studies. Until a counter-example to the trend given in the current paper is shown, the findings of this work are useful and generally applicable.
> >
> > Consequently, I maintain my score.

---

> > > ### Author Response · Authors · 2025-04-07
> > >
> > > Thank you for your encouraging feedback. Regarding the $(1 + \epsilon R^{\gamma})$ term, our approach aligns with previous studies on scaling laws, as referenced in [1]. We base our work on the trends in losses observed across different model variants. The proposed scaling law is linked to model shape, particularly through the relationship with $R$. Therefore, $R$ would be one part of scaling laws. Furthermore, we introduce two learnable parameters, $\epsilon$ and $\gamma$, to refine the smoothness of the scaling law. If you have follow-up questions, please let us know! Thanks!
> > >
> > > [1] Gadre, Samir Yitzhak, et al. "Language models scale reliably with over-training and on downstream tasks." arXiv preprint arXiv:2403.08540 (2024).

---

### Official Review · Reviewer_zGHh · 2025-03-14

**Overall Recommendation:** 4

**Summary:**

The paper presents revised inference-time scaling laws based on model architecture choices, relying on the observation that models of the same size but different architecture choices can have up to a 3.5 times difference in inference latency. Using that, they train models of varying sizes up to 1B parameters, improving inference latency by a factor of 1.8 while keeping the same accuracy on downstream tasks as that of open-source models of the same size.

**Claims And Evidence:**

Claims are supported by extensive experiments and ablations.

**Essential References Not Discussed:**

None to be discussed to the best of my knowledge.

**Experimental Designs Or Analyses:**

Experimental settings are methodologically sound, experiments and ablations are informative and extensive. See questions for aspects to clarify.

**Methods And Evaluation Criteria:**

Authors use standard evaluations, benchmark datasets and baselines which are sound.

**Other Comments Or Suggestions:**

Nothing particularly important to mention.

**Other Strengths And Weaknesses:**

None, all submitted either as part of the paper evaluation or as questions.

**Questions For Authors:**

- Table 1 seems to only rely on 6 data points, how statistically valid is extrapolating to larger sizes in this case?
- The Spearman correlation analysis (Figure 7c) strongly supports the scaling law’s ranking capability, could you explain the negative correlation for Chinchilla’s law?
- The modified scaling law in Eq. 4 introduces a $(1 + \epsilon R^\gamma)$ term to the Chinchilla loss function, while it works empirically, is there any theoretical rationale behind that?

**Relation To Broader Scientific Literature:**

The paper is relevant as a training recipe for future generations of LLMs and shows a novel finding relating inference latency with architecture design choices, which motivates further work in this direction.

**Theoretical Claims:**

Nothing to discuss in particular.

---

> ### Author Rebuttal · Authors · 2025-03-31
>
> **Q1:** Table 1 seems to only rely on 6 data points, how statistically valid is extrapolating to larger sizes in this case?
>
> **A1:** As detailed in Table 4 of the Appendix, each model size includes several variants. We use 27 data points to fit the scaling laws in Figure 7.
>
> **Q2:** The Spearman correlation analysis (Figure 7c) strongly supports the scaling law’s ranking capability. Could you explain the negative correlation for Chinchilla’s law?
>
> **A2:** The Spearman correlation ranges from -1 to 1. In Figure 7, the actual rank of 4 models are [1, 2, 4, 3], however, the predicted rank of Chinchilla’s law is [4, 2, 3, 1]. Therefore, the value of the Spearman correlation analysis is negative.
>
> **Q3:** The modified scaling law in Eq. 4 introduces a $(1 + \epsilon R^\gamma)$ term to the Chinchilla loss function. While it works empirically, is there any theoretical rationale behind that?
>
> **A3:** Similar to prior work [1] on scaling laws, our work is guided by the trend of losses observed across various model variants.
>
> [1] Gadre, Samir Yitzhak, et al. "Language models scale reliably with over-training and on downstream tasks." arXiv preprint arXiv:2403.08540 (2024).

---

### Official Review · Reviewer_B9RC · 2025-03-14

**Overall Recommendation:** 1

**Summary:**

Traditional scaling laws (like Chinchilla) do not account for model architectures in their modeling of the loss. This paper first highlights that the model architecture (like hidden dim, #layers) affects the downstream loss as well as the latency of the models (also studied in multiple previous works). They propose incorporating model shape (width to depth ratio) in the chinchilla scaling law. They use this scaling law to extract a couple of choices for aspect ratios that might give a good enough (top-k) loss. The inference latency of these choices are then evaluated empirically, and combined with the loss scores/ranks to guide the final choice of model to train.

**Claims And Evidence:**

Upto some extent (See weakness section for details)

**Essential References Not Discussed:**

Some very relevant missing references:
https://arxiv.org/pdf/2109.10686 (SCALE EFFICIENTLY: INSIGHTS FROM PRE-TRAINING AND FINE-TUNING TRANSFORMERS) that talks about impact of model shape on downstream loss
https://arxiv.org/pdf/2305.13035 (effect of model shape in vision tasks)

**Experimental Designs Or Analyses:**

Upto some extent (See weakness section for details)

**Methods And Evaluation Criteria:**

Yes

**Other Comments Or Suggestions:**

See the weakness section

**Other Strengths And Weaknesses:**

### Strength:
- I really liked authors using their scaling law observations to come up with the most efficient (in terms of latency) 1B model.


### Weaknesses:

- The authors emphasize the importance of estimating the downstream inference latency of models. However, they still propose to estimate model latency using an untrained model. In their final model selection algorithm, the latency scores and the predicted loss are combined to select k candidates for training. It would have been much more interesting if the scaling law could be formulated to directly predict a combined metric of inference latency and loss, rather than treating them separately. Can we not estimate inference latency based on d/n? Wondering what were the reasons behind author's choice of the specific modeling they present in the paper.

- Moreover, inference latency depends on the number of tokens the model generates to answer a question at inference. More accurate models might be able to answer the same question in fewer tokens, through higher-level reasoning or by reducing unnecessary attempts. Given these factors, predicting inference latency solely based on architecture (using an untrained model), as the authors propose, seems problematic.

- I strongly encourage the authors to precisely define how they are estimating downstream latency.

- The authors incorporate the d/n term in their modified scaling law. While Figure 7 shows that this scaling law better estimates loss for the models they trained, my main concern is: how much does the d/n term vary in real-world models? In Figure 4, they show loss variation as d/n ranges from \(2^2\) to \(2^9\), but does it vary this much in actual models that practitioners train (e.g., Gemma, LLaMA, Qwen series)? I encourage the authors to demonstrate how much the d/n term (the ratio R) varies across existing open-source models and how accurately one could use it to estimate their loss. They could, for instance, use a best-fit approximation for a held-out dataset to estimate the loss (perplexity) of these models or, at the very least, highlight the actual range of variation of R across these models.

- This concern is further supported by Table 2. Among the top three Morph-1B model variants trained by the authors, while the aspect ratio varies significantly, the downstream performance remains nearly the same. This suggests that for any **reasonable range of choice** of aspect ratio, d/n may not be that critical in modeling loss through scaling laws.

- Finally, the prediction accuracy of the trained scaling laws has only been evaluated on models 3× larger than those in the training set. In general, one would want to test at least an order of magnitude larger models to validate the law’s extrapolation capabilities. However, I understand  this might stem out of computational constraints, and I will not weigh this limitation heavily in my final assessment.

### Missing References:
The paper overlooks some relevant prior work that studies the impact of model shape on scaling laws:
- [Scale Efficiently: Insights from Pre-training and Fine-tuning Transformers (Tay et al., 2022)](https://arxiv.org/pdf/2109.10686) – Discusses how model shape (depth vs. width) impacts downstream performance and questions whether existing scaling laws capture these effects.
- [Effect of Model Shape in Vision Tasks (Alabdulmohsin et al., 2023)](https://arxiv.org/pdf/2305.13035#:~:text=Unfortunately%2C%20in%20both%20,up) – Explores how model width and depth interact in Vision Transformers (ViTs) and whether similar trends hold across domains.

**Questions For Authors:**

See the weakness section

**Relation To Broader Scientific Literature:**

Helps guide choice of model architecture to train for practioners.

**Theoretical Claims:**

NA

---

> ### Author Rebuttal · Authors · 2025-03-31
>
> **Q1:** The authors emphasize the importance of estimating the downstream inference latency of models...
>
> **Q2:** Moreover, inference latency depends on the number of tokens the model generates to answer a question at inference...
>
> **Q3:** I strongly encourage the authors to define precisely how they are estimating downstream latency.
>
> **A1 & A2, & A3:** We believe these three questions are related and address them together. Firstly, we agree with the reviewer that the end-to-end latency for downstream tasks depends on the number of tokens generated to answer the questions. End-to-end latency can be seen as the time per output token times the number of generated tokens. However, predicting the number of tokens generated to answer downstream tasks is challenging since we do not target any specific downstream task during pre-training. Therefore, we follow the approach used in previous work [1, 2]; we measure the latency of models by fixing the number of input and output tokens and focus on minimizing the time per output token. For example, we fix the input length as 128 and the output length as 256 in Figure 1. As a result, we don't need to model the latency in the scaling law and can instead measure a model's time per output token using an untrained model. Thus, we use empirically measured model latencies in our proposed model selection pipeline.
>
> We would also like to stress that reducing response length while maintaining answer quality is of independent research interest and orthogonal to our work. In our paper, we follow the previous work [1, 2] and measure the latency of models while fixing the number of input and output tokens. We will make it clearer in the main text in the final version.
>
> **Q4:** The authors incorporate the d/n term in their modified scaling law....
>
> **A4:** Here, we show the d/n term for 15 open-sourced models: https://anonymous.4open.science/r/ICML25-Rebuttal-3B34/open-sourced-models-d_n.png
>
> In Figure 4, we illustrate the variation in loss as d/n ranges from (2^4) to (2^9) rather than from (2^2) to (2^9). From the above table, we believe that our ranges are reasonable.
>
> **Q5:** This concern is further supported by Table 2. Among the top three Morph-1B model variants trained by the authors, while the aspect ratio varies significantly....
>
> **A5:** Our experimental results indicate that an aspect ratio between 64 and 256 is a practical choice for 1B models. However, as illustrated in Figure 5, the performance of two models on downstream tasks can differ even if their losses are similar. For instance, while the losses of two 164M model variants are recorded as 3.32 and 3.35, their accuracies on the BoolQ dataset differ significantly, with scores of 0.5379 and 0.5734, respectively. Therefore, while the d/n ratio may have a limited impact on model loss under a reasonable range, the performance on downstream tasks varies.
>
> **Q6:** Finally, the prediction accuracy of the trained scaling laws has only been evaluated on models 3× larger than those in the training set....
>
> **A6:** Thanks for your suggestions! We agree with your points. We plan to expand our experiments to include larger models when we obtain more computational resources.
>
> Missing References:
> The paper overlooks some relevant prior work that studies the impact of model shape on scaling laws:
> Scale Efficiently: Insights from Pre-training and Fine-tuning Transformers (Tay et al., 2022) – Discusses how model shape (depth vs. width) impacts downstream performance and questions whether existing scaling laws capture these effects.
> Effect of Model Shape in Vision Tasks (Alabdulmohsin et al., 2023) – Explores how model width and depth interact in Vision Transformers (ViTs) and whether similar trends hold across domains.
>
> Thank you for the suggestions. The 'Scale Efficiently' paper explores the influence of model shape on downstream performance, but it ignores the effect on inference efficiency. Regarding the second paper, 'Effect of Model Shape in Vision Tasks,' it focuses on the impact of model shape on Vision Transformers (ViTs), whereas our study examines its impact on Large Language Models (LLMs). We will include these papers in the related work section.
>
> [1] Zhong, Yinmin, et al. "{DistServe}: Disaggregating prefill and decoding for goodput-optimized large language model serving." OSDI 24.
>
> [2] Yu, Gyeong-In, et al. "Orca: A distributed serving system for {Transformer-Based} generative models." OSDI 22.

---

> > ### Comment · Reviewer_B9RC · 2025-04-04
> >
> > I thank the authors for their response.
> >
> > > Range of d/n ratio for open-source models
> >
> > From the d/n ratios of open-source models linked in the response, it is evident that d/n typically falls within the range of 2⁶ to 2⁷. In contrast, the authors argue for searching across a much wider range, which does not appear practical. For instance, the authors' suggested optimal d/n value is far outside the commonly observed range.
> >
> > > BoolQ dataset results
> >
> > The reported scores on the BoolQ dataset (0.5379 vs. 0.5734) are quite low for a binary classification task. A difference between 53% and 57% accuracy is not sufficient to draw strong conclusions. Therefore, my concern remains that the choice of d/n may not have a meaningful impact on downstream performance, especially when compared to a natural baseline guess informed by existing models.
> >
> > I once again thank the authors for their response. However, I believe the manuscript still requires substantial improvement in justifying the search for optimal d/n and showing that it offers real benefits over conventional heuristics. My concerns remain, and I am maintaining my current rating.

---

> > > ### Author Response · Authors · 2025-04-05
> > >
> > > Thanks for your response! Our answers are as follows:
> > >
> > > > Range of d/n ratio for open-source models
> > >
> > > First, we present additional results for the d/n term from open-sourced models here: https://anonymous.4open.science/r/ICML25-Rebuttal-3B34/open-sourced-models-d_n-rebuttal.png. From https://anonymous.4open.science/r/ICML25-Rebuttal-3B34/open-sourced-models-d_n.png and https://anonymous.4open.science/r/ICML25-Rebuttal-3B34/open-sourced-models-d_n-rebuttal.png, our observation shows that the minimum d_n value among open-sourced models is 29.54 (MiniCPM-1B [1]), and the maximum is 199.11 (Gemma-3-27B [2]), which supports the d/n range we discuss in our paper as being reasonable. Furthermore, with 21 unique d_n values identified, a comprehensive search range is required. Since pre-training requires significant time and resources, developing effective scaling laws based on model shape could help inform model design decisions and reduce training costs.
> > >
> > > > BoolQ dataset results
> > >
> > > First, we do not believe that the reported scores on the BoolQ dataset are quite low for two 164M model variants compared with the results reported in https://paperswithcode.com/sota/question-answering-on-boolq. Moreover, we present more results of the 1B model here: while the losses of Morph-1B-v1 and Morph-1B-v2 are recorded as 2.896 and 2.909, their accuracies on the BoolQ dataset differ significantly, with scores of 0.5758 and 0.6049, respectively. Given OPT-IML 1.3B (zero-shot)'s performance of 61.5% on BoolQ, as documented at https://paperswithcode.com/sota/question-answering-on-boolq, we believe our results are strong for our class of models even with much fewer training resources and demonstrate meaningful differentiation in downstream performance.
> > >
> > > [1] Hu, Shengding, et al. "Minicpm: Unveiling the potential of small language models with scalable training strategies." arXiv preprint arXiv:2404.06395 (2024).
> > >
> > > [2] Team, Gemma, et al. "Gemma 3 Technical Report." arXiv preprint arXiv:2503.19786 (2025).

---

### Decision · Program_Chairs · 2025-05-01

**Decision:**

Accept (poster)

**Comment:**

The paper modifies the training-compute optimal (Chinchilla) scaling laws to take inference costs into accounts. The paper provides extensive experiments to validate the proposed scaling laws for inference-efficient models.

The paper got very mixed reviews:
- Reviewer B9RC (1: reject) has a number of concerns:
1. Prediction accuracy is only measured by training a 3x larger model.
2. Has concerns about d/n term, specifically how much it matters and how different it is in practical models. The reviewer points out that the range of d/n in practical models is smaller than that considered for the search by the paper (30-200 vs 16-512), which is true, but the range considered in the paper is somewhat reasonable.
3. The reviewer has concerns about how the inference cost is measured, and notes that inference latency depends on the number of tokens a model generates as well. The authors measure this in tokens, not taking the number of tokens generated to a query into account, and this is standard practice and reasonable.
- Reviewer (4: accept) has no concerns and raised no major point in either direction.
- Reviewer 4YJ4 (5: strong accept) emphasized that the empirical procedure is sound and that the results are of practical value. The reviewer also emphasizes that the primary claims are supported with clear explanations and evidence.
- Reviewer kVaw (2: weak reject) notes (like reviewer B9RC) that the aspect ratio (d/n) seems to have a relatively minor effect on performance, and notes that to only characterize the model architecture by d/n is a limitation, in particular since the claim is relatively broad (i.e., the authors talk broadly about inference-efficient models). I agree with the reviewer, and recommend the authors explain early on that they focus on transformers and only vary the ratio d/n. However, taking d/n into account in scaling laws is interesting, even if the finding is that the impact is not huge.

While the paper has the weaknesses mentioned above, I agree with reviewer 4YJ4 (also based on my own reading) that the paper sound and the conclusions are interesting. I agree with some of the limitations raised by reviewer B9RC and kVaw, and think that they can be addressed in the final version by making a bit more narrow and specific claims.